# Abide by the Law and Follow the Flow: Conservation Laws for Gradient Flows

**Sibylle Marcotte**
ENS - PSL Univ.
sibylle.marcotte@ens.fr

**Rémi Gribonval**
Univ Lyon, EnsL, UCBL,
CNRS, Inria, LIP,
remi.gribonval@inria.fr

**Gabriel Peyré**
CNRS, ENS - PSL Univ.
gabriel.peyre@ens.fr

## Abstract

Understanding the geometric properties of gradient descent dynamics is a key ingredient in deciphering the recent success of very large machine learning models. A striking observation is that trained over-parameterized models retain some properties of the optimization initialization. This "implicit bias" is believed to be responsible for some favorable properties of the trained models and could explain their good generalization properties. The purpose of this article is threefold. First, we rigorously expose the definition and basic properties of "conservation laws", that define quantities conserved during gradient flows of a given model (e.g. of a ReLU network with a given architecture) with any training data and any loss. Then we explain how to find the maximal number of independent conservation laws by performing finite-dimensional algebraic manipulations on the Lie algebra generated by the Jacobian of the model. Finally, we provide algorithms to: a) compute a family of polynomial laws; b) compute the maximal number of (not necessarily polynomial) independent conservation laws. We provide showcase examples that we fully work out theoretically. Besides, applying the two algorithms confirms for a number of ReLU network architectures that all known laws are recovered by the algorithm, and that there are no other independent laws. Such computational tools pave the way to understanding desirable properties of optimization initialization in large machine learning models.

## 1 Introduction

State-of-the-art approaches in machine learning rely on the conjunction of gradient-based optimization with vastly "over-parameterized" architectures. A large body of empirical [30] and theoretical [5] works suggest that, despite the ability of these models to almost interpolate the input data, they are still able to generalize well. Analyzing the training dynamics of these models is thus crucial to gain a better understanding of this phenomenon. Of particular interest is to understand what properties of the initialization are preserved during the dynamics, which is often loosely referred to as being an "implicit bias" of the training algorithm. The goal of this article is to make this statement precise, by properly defining maximal sets of such "conservation laws", by linking these quantities to algebraic computations (namely a Lie algebra) associated with the model parameterization (in our framework, this parameterization is embodied by a re-parameterization mapping $\phi$), and finally by exhibiting algorithms to implement these computations in SageMath [29].

**Over-parameterized model**  Modern machine learning practitioners and researchers have found that over-parameterized neural networks (with more parameters than training data points), which are often trained until perfect interpolation, have impressive generalization properties [30, 5]. This performance seemingly contradicts classical learning theory [25], and a large part of the theoretical deep learning literature aims at explaining this puzzle. The choice of the optimization algorithm is crucial to the model generalization performance [10, 21, 14], thus inducing an *implicit bias*.

37th Conference on Neural Information Processing Systems (NeurIPS 2023).

**Implicit bias**   The terminology "implicit bias" informally refers to properties of trained models which are induced by the optimization procedure, typically some form of regularization [22]. For gradient descent, in simple cases such as scalar linear neural networks or two-layer networks with a single neuron, it is actually possible to compute in closed form the implicit bias, which induces some approximate or exact sparsity regularization [10]. Another interesting case is logistic classification on separable data, where the implicit bias selects the max-margin classifier both for linear models [26] and for two-layer neural networks in the mean-field limit [8]. A key hypothesis to explicit the implicit bias is often that the Riemannian metric associated to the over-parameterization is either of Hessian type [10, 3], or can be somehow converted to be of Hessian type [3], which is seemingly always a very strong constraint. For example, even for simple two-layer linear models (i.e., matrix factorization) with more than a single hidden neuron, the Hessian type assumption does not hold, and no closed form is known for the implicit bias [11]. The work of [17] gives conditions on the over-parameterization for this to be possible (for instance certain Lie brackets should vanish). These conditions are (as could be expected) stronger than those required to apply Frobenius theory, as we do in the present work to retrieve conservation laws.

**Conservation laws**   Finding functions conserved during gradient flow optimization of neural networks (a continuous limit of gradient descent often used to model the optimization dynamics) is particularly useful to better understand the flow behavior. One can see conservation laws as a "weak" form of implicit bias: to explain, among a possibly infinite set of minimizers, which properties (e.g. in terms of sparsity, low-rank, etc.) are being favored by the dynamic. If there are enough conservation laws, one has an exact description of the dynamic (see Section 3.4), and in some cases, one can even determine explicitly the implicit bias. Otherwise, one can still predict what properties of the initialization are retained at convergence, and possibly leverage this knowledge. For example, in the case of linear neural networks, certain *balancedness properties* are satisfied and provide a class of conserved functions [24, 9, 1, 2, 15, 28, 19]. These conservation laws enable for instance to prove the global convergence of the gradient flow [4] under some assumptions. We detail these laws in Proposition 4.1. A subset of these "balancedness" laws still holds in the case of a ReLU activation [9], which reflects the rescaling invariance of these networks (see Section 4 for more details). More generally such conservation laws bear connections with the invariances of the model [16]: to each 1-parameter group of transformation preserving the loss, one can associate a conserved quantity, which is in some sense analogous to Noether's theorem [23, 12]. Similar reasoning is used by [31] to show the influence of initialization on convergence and generalization performance of the neural network. Our work is somehow complementary to this line of research: instead of assuming a priori known symmetries, we directly analyze the model and give access to conservation laws using algebraic computations. For matrix factorization as well as for certain ReLU network architectures, this allows us to show that the conservation laws reported in the literature are complete (there are no other independent quantities that would be preserved by all gradient flows).

## Contributions

We formalize the notion of a conservation law, a quantity preserved through all gradient flows given a model architecture (e.g. a ReLU neural network with prescribed layers). Our main contributions are:

- to show that for several classical losses, characterizing conservation laws for deep linear (resp. shallow ReLU) networks boils down to analyzing a finite dimensional space of vector fields;

- to propose an algorithm (coded in SageMath) identifying polynomial conservation laws on linear / ReLU network architectures; it identifies all known laws on selected examples;

- to formally define the maximum number of (not necessarily polynomial) independent conservation laws and characterize it a) theoretically via Lie algebra computations; and b) practically via an algorithm (coded in SageMath) computing this number on worked examples;

- to illustrate that in certain settings these findings allow to rewrite an over-parameterized flow as an "intrinsic" low-dimensional flow;

- to highlight that the cost function associated to the training of linear and ReLU networks, shallow or deep, with various losses (quadratic and more) fully fits the proposed framework.

A consequence of our results is to show for the first time that conservation laws commonly reported in the literature are maximal: there is no other independent preserved quantity (see Propositions 4.3, 4.2, Corollary 4.4) and Section 4.2).

## 2 Conservation Laws for Gradient Flows

After some reminders on gradient flows, we formalize the notion of conservation laws.

### 2.1 Gradient dynamics

We consider learning problems, where we denote $x_i \in \mathbb{R}^m$ the features and $y_i \in \mathcal{Y}$ the targets (for regression, typically with $\mathcal{Y} = \mathbb{R}^n$) or labels (for classification) in the case of supervised learning, while $y_i$ can be considered constant for unsupervised/self-supervised learning. We denote $X \coloneqq (x_i)_i$ and $Y \coloneqq (y_i)_i$. Prediction is performed by a parametric mapping $g(\theta, \cdot) : \mathbb{R}^m \to \mathbb{R}^n$ (for instance a neural network) which is trained by empirically minimizing over parameters $\theta \in \Theta \subseteq \mathbb{R}^D$ a **cost**

$$\mathcal{E}_{X,Y}(\theta) \coloneqq \sum_i \ell(g(\theta, x_i), y_i), \tag{1}$$

where $\ell$ is the **loss** function. In practical examples with linear or ReLU networks, $\Theta$ is either $\mathbb{R}^D$ or an open set of "non-degenerate" parameters. The goal of this paper is to analyze what functions $h(\theta)$ are preserved during the gradient flow (the continuous time limit of gradient descent) of $\mathcal{E}_{X,Y}$:

$$\dot{\theta}(t) = -\nabla \mathcal{E}_{X,Y}(\theta(t)), \text{ with } \theta(0) = \theta_{\text{init}}. \tag{2}$$

A priori, one can consider different "levels" of conservation, depending whether $h$ is conserved: during the optimization of $\mathcal{E}_{X,Y}$ for a given loss $\ell$ and a given data set $(x_i, y_i)_i$; or given a loss $\ell$, during the optimization of $\mathcal{E}_{X,Y}$ for *any* data set $(x_i, y_i)_i$. Note that using stochastic optimization methods and discrete gradients would break the exact preservation of the conservation laws, and only approximate conservation would hold, as remarked in [16].

### 2.2 Conserved functions

As they are based on gradient flows, conserved functions are first defined locally.

**Definition 2.1** (Conservation through a flow). Consider an open subset $\Omega \subseteq \Theta$ and a vector field $\chi \in \mathcal{C}^1(\Omega, \mathbb{R}^D)$. By the Cauchy-Lipschitz theorem, for each initial condition $\theta_{\text{init}}$, there exists a unique maximal solution $t \in [0, T_{\theta_{\text{init}}}) \mapsto \theta(t, \theta_{\text{init}})$ of the ODE $\dot{\theta}(t) = \chi(\theta(t))$ with $\theta(0) = \theta_{\text{init}}$. A function $h : \Omega \subseteq \mathbb{R}^D \to \mathbb{R}$ is *conserved on $\Omega$ through the vector field $\chi$* if $h(\theta(t, \theta_{\text{init}})) = h(\theta_{\text{init}})$ for each choice of $\theta_{\text{init}} \in \Omega$ and every $t \in [0, T_{\theta_{\text{init}}})$. It is *conserved on $\Omega$ through a subset $W \subset \mathcal{C}^1(\Omega, \mathbb{R}^D)$* if $h$ is conserved on $\Omega$ during all flows induced by all $\chi \in W$.

In particular, one can adapt this definition for the flow induced by the cost (1).

**Definition 2.2** (Conservation during the flow (2) with a given dataset). Consider an open subset $\Omega \subseteq \Theta$ and a dataset $(X, Y)$ such that $\mathcal{E}_{X,Y} \in \mathcal{C}^2(\Omega, \mathbb{R})$. A function $h : \Omega \subseteq \mathbb{R}^D \to \mathbb{R}$ is *conserved on $\Omega$ during the flow* (2) if it is conserved through the vector field $\chi(\cdot) \coloneqq \nabla \mathcal{E}_{X,Y}(\cdot)$.

Our goal is to study which functions are conserved during *"all" flows* defined by the ODE (2). This in turn leads to the following definition.

**Definition 2.3** (Conservation during the flow (2) with "any" dataset). Consider an open subset $\Omega \subset \Theta$ and a loss $\ell(z, y)$ such that $\ell(\cdot, y)$ is $\mathcal{C}^2$-differentiable for all $y \in \mathcal{Y}$. A function $h : \Omega \subseteq \mathbb{R}^D \to \mathbb{R}$ is *conserved on $\Omega$ for any data set* if, for each data set $(X, Y)$ *such that* $g(\cdot, x_i) \in \mathcal{C}^2(\Omega, \mathbb{R})$ for each $i$, the function $h$ is conserved on $\Omega$ during the flow (2). This leads us to introduce the family of vector fields:

$$W_\Omega^g \coloneqq \left\{ \chi(\cdot) : \exists X, Y, \forall i \; g(\cdot, x_i) \in \mathcal{C}^2(\Omega, \mathbb{R}), \; \chi = \nabla \mathcal{E}_{X,Y} \right\} \subseteq \mathcal{C}^1(\Omega, \mathbb{R}^D) \tag{3}$$

so that being conserved on $\Omega$ for any dataset is the same as being conserved on $\Omega$ through $W_\Omega^g$.

The above definitions are local and conditioned on a choice of open set of parameters $\Omega \subset \Theta$. We are rather interested in functions defined on the whole parameter space $\Theta$, hence the following definition.

**Definition 2.4.** A function $h : \Theta \mapsto \mathbb{R}$ is *locally conserved on $\Theta$ for any data set* if for each open subset $\Omega \subseteq \Theta$, $h$ is conserved on $\Omega$ for any data set.

A basic property of $\mathcal{C}^1$ conserved functions (which proof can be found in Appendix A) corresponds to an "orthogonality" between their gradient and the considered vector fields.

**Proposition 2.5.** *Given a subset $W \subset \mathcal{C}^1(\Omega, \mathbb{R}^D)$, its* trace *at $\theta \in \Omega$ is defined as the linear space*

$$W(\theta) := \mathrm{span}\{\chi(\theta) : \chi \in W\} \subseteq \mathbb{R}^D. \tag{4}$$

*A function $h \in \mathcal{C}^1(\Omega, \mathbb{R})$ is conserved on $\Omega$ through $W$ if, and only if $\nabla h(\theta) \perp W(\theta), \forall \theta \in \Omega$.*

Therefore, combining Proposition 2.5 and Definition 2.4, the object of interest to study locally conserved functions is the union of the traces

$$W_\theta^g := \bigcup \left\{ W_\Omega^g(\theta) \ : \ \Omega \subseteq \Theta \text{ with } \Omega \text{ a neighborhood of } \theta \right\}. \tag{5}$$

**Corollary 2.6.** *A function $h : \Theta \mapsto \mathbb{R}$ is locally conserved on $\Theta$ for any data set if and only if $\nabla h(\theta) \perp W_\theta^g$ for all $\theta \in \Theta$.*

It will soon be shown (cf Theorem 2.14) that $W_\theta^g$ can be rewritten as the trace $W(\theta)$ of a *simple* finite-dimensional functional space $W$. Meanwhile, we keep the specific notation. For the moment, this set is explicitly characterized via the following proposition (which proof can be found in Appendix B).

**Proposition 2.7.** *Assume that for each $y \in \mathcal{Y}$ the loss $\ell(z, y)$ is $\mathcal{C}^2$-differentiable with respect to $z \in \mathbb{R}^n$. For each $\theta \in \Theta$ we have:*

$$W_\theta^g = \mathop{\mathrm{span}}_{(x,y) \in \mathcal{X}_\theta \times \mathcal{Y}} \{[\partial_\theta g(\theta, x)]^\top \nabla_z \ell(g(\theta, x), y)\}$$

*where $\mathcal{X}_\theta$ is the set of data points $x$ such that $g(\cdot, x)$ is $\mathcal{C}^2$-differentiable in the neighborhood of $\theta$.*

*Example* 2.8. As a first simple example, consider a two-layer *linear* neural network in dimension 1 (both for the input and output), with a single neuron. For such – admittedly trivial – architecture, the parameter is $\theta = (u, v) \subseteq \mathbb{R}^2$ and the model writes $g(\theta, x) = uvx$. One can directly check that the function: $h(u, v) = u^2 - v^2$ is *locally conserved on $\mathbb{R}^2$ for any data set*. Indeed in that case $\nabla h(u, v) = (2u, -2v)^\top \perp W_\theta^g = \mathop{\mathrm{span}}_{(x,y) \in \mathbb{R} \times \mathcal{Y}} \{(vx, ux)^\top \nabla_z \ell(g(\theta, x), y)\} = \mathbb{R} \times (v, u)^\top$ given that the gradient $\nabla_z \ell(g(\theta, x), y)$ is an arbitrary scalar.

In this example we obtain a simple expression of $W_\theta^g$, however in general cases it is not possible to obtain such a simple expression from Proposition 2.7. We will show that in some cases, it is possible to express $W_\theta^g$ as the trace $W(\theta)$ of a simple finite-dimensional space $W$ (cf. Theorem 2.14).

### 2.3 Reparametrization

To make the mathematical analysis tractable and provide an algorithmic procedure to determine these functions, our fundamental hypothesis is that the **model** $g(\theta, x)$ can be (locally) factored via a **reparametrization** $\phi$ as $f(\phi(\theta), x)$. We require that the model $g(\theta, x)$ satisfies the following central assumption.

**Assumption 2.9** (Local reparameterization). There exists $d$ and $\phi \in \mathcal{C}^2(\Theta, \mathbb{R}^d)$ such that: for each parameter $\theta_0$ in the open set $\Theta \subseteq \mathbb{R}^D$, for each $x \in \mathcal{X}$ such that $\theta \mapsto g(\theta, x)$ is $\mathcal{C}^2$ in a neighborhood of $\theta_0$[1], there is a neighborhood $\Omega$ of $\theta_0$ and $f(\cdot, x) \in \mathcal{C}^2(\phi(\Omega), \mathbb{R}^n)$ such that

$$\forall \theta \in \Omega, \quad g(\theta, x) = f(\phi(\theta), x). \tag{6}$$

Note that if the model $g(\cdot, x)$ is smooth on $\Omega$ then (6) is always satisfied with $\phi := \mathtt{id}$ and $f(\cdot, x) := g(\cdot, x)$, yet this trivial factorization fails to capture the existence and number of conservation laws as studied in this paper. This suggests that, among all factorizations shaped as (6), there may be a notion of an optimal one.

*Example* 2.10. (Factorization for *linear* neural networks) In the two-layer case, with $r$ neurons, denoting $\theta = (U, V) \in \mathbb{R}^{n \times r} \times \mathbb{R}^{m \times r}$ (so that $D = (n + m)r$), we can factorize $g(\theta, x) := UV^\top x$ by the reparametrization $\phi(\theta) := UV^\top \in \mathbb{R}^{n \times m}$ using $f(\phi, x) = \phi \cdot x$. More generally for $q$ layers, with $\theta = (U_1, \cdots, U_q)$, we can still factorize $g(\theta, x) := U_1 \cdots U_q x$ using $\phi(\theta) := U_1 \cdots U_q$ and the same $f$. This factorization is *globally* valid on $\Omega = \Theta = \mathbb{R}^D$ since $f(\cdot, x)$ does not depend on $\theta_0$.

The notion of locality of the factorization (6) is illustrated by the next example.

---

[1]i.e., $x$ belongs to the set $\mathcal{X}_{\theta_0}$, as defined in Proposition 2.7.

*Example* 2.11 (Factorization for two-layer ReLU networks). Consider $g(\theta, x) = \left( \sum_{j=1}^r u_{k,j} \sigma(\langle v_j, x \rangle + b_j) + c_k \right)_{k=1}^n$, with $\sigma(t) := \max(t, 0)$ the ReLU activation function and $v_j \in \mathbb{R}^m$, $u_{k,j} \in \mathbb{R}$, $b_j, c_k \in \mathbb{R}$. Then, denoting $\theta = (U, V, b, c)$ with $U = (u_{k,j})_{k,j} =: (u_1, \cdots, u_r) \in \mathbb{R}^{n \times r}$, $V = (v_1, \cdots, v_r) \in \mathbb{R}^{m \times r}$, $b = (b_1, \cdots, b_r)^\top \in \mathbb{R}^r$ and $c = (c_1, \cdots, c_n) \in \mathbb{R}^n$ (so that $D = (n + m + 1)r + n$), we rewrite $g(\theta, x) = \sum_{j=1}^r u_j \varepsilon_{j,x} \left( v_j^\top x + b_j \right) + c$ where, given $x$, $\varepsilon_{j,x} = \mathbb{1}(v_j^\top x + b_j > 0)$ is piecewise constant with respect to $\theta$. Consider $\theta^0 = (U^0, V^0, b^0, c^0) \in \mathbb{R}^D$ where $V^0 = (v_1^0, \cdots, v_r^0)$ and $b^0 = (b_1^0, \cdots, b_r^0)^\top$. Then the set $\mathcal{X}_{\theta^0}$ introduced in Proposition 2.7 is $\mathcal{X}_{\theta^0} = \mathbb{R}^m - \cup_j \{v_j^{0\top} x + b_j^0 = 0\}$. Let $x \in \mathcal{X}_{\theta^0}$. Then on any domain $\Omega \subset \mathbb{R}^D$ such that $\theta^0 \in \Omega$ and $\varepsilon_{j,x}(\theta) := \mathbb{1}(v_j^\top x + b_j > 0)$ is constant over $\theta \in \Omega$, the model $g_\theta(x)$ can be factorized by the reparametrization $\phi(\theta) = ((u_j v_j^\top, u_j b_j)_{j=1}^r, c)$. In particular, in the case without bias $((b, c) = (0, 0))$, the reparametrization is defined by $\phi(\theta) = (\phi_j)_{j=1}^r$ where $\phi_j = \phi_j(\theta) := u_j v_j^\top \in \mathbb{R}^{n \times m}$ (here $d = rmn$) using $f(\phi, x) = \sum_j \varepsilon_{j,x} \phi_j x$: the reparametrization $\phi(\theta)$ contains $r$ matrices of size $m \times n$ (each of rank at most one) associated to a "local" $f(\cdot, x)$ valid in a neighborhood of $\theta$. A similar factorization is possible for deeper ReLU networks [27], as further discussed in the proof of Theorem 2.14 in Appendix C.

Combining Proposition 2.7 and using chain rules, we get a new characterization of $W_\theta^g$:

**Proposition 2.12.** *Assume that the loss $\ell(z, y)$ is $\mathcal{C}^2$-differentiable with respect to $z$. We recall (cf (5)) that $W_\theta^g := \cup_{\Omega \subseteq \Theta: \Omega \text{ is open and } \Omega \ni \theta} W_\Omega^g(\theta)$. Under Assumption 2.9, for all $\theta \in \Theta$:*

$$W_\theta^g = \partial\phi(\theta)^\top W_{\phi(\theta)}^f \tag{7}$$

*with $\partial\phi(\theta) \in \mathbb{R}^{d \times D}$ the Jacobian of $\phi$ and $W_{\phi(\theta)}^f := \operatorname*{span}_{(x,y) \in \mathcal{X}_\theta \times \mathcal{Y}} \{\partial f^x(\phi(\theta))^\top \nabla_z \ell(g(\theta, x), y)\}$,*

*where $f^x(\cdot) := f(\cdot, x)$.*

We show in Section 2.4 that, under mild assumptions on the loss $\ell$, $W_{\phi(\theta)}^f = \mathbb{R}^d$, so that Proposition 2.12 yields $W_\theta^g = \operatorname{range}(\partial\phi(\theta)^\top)$. Then by Corollary 2.6, a function $h$ that is locally conserved on $\Theta$ for any data set is *entirely characterized* via the kernel of $\partial\phi(\theta)^\top$: $\partial\phi(\theta)^\top \nabla h(\theta) = 0$ for all $\theta \in \Theta$. The core of our analysis is then to analyze the (Lie algebraic) structure of $\operatorname{range}(\partial\phi(\cdot)^\top)$.

## 2.4 From conserved functions to conservation laws

For linear and ReLU networks we show in Theorem 2.14 and Proposition 2.16 that under (mild) assumptions on the loss $\ell(\cdot, \cdot)$, being locally conserved on $\Theta$ for any data set (according to Definition 2.4) is the same as being conserved (according to Definition 2.1) on $\Theta$ through the *finite-dimensional* subspace

$$W_\phi := \operatorname{span}\{\nabla\phi_1(\cdot), \cdots, \nabla\phi_d(\cdot)\} = \left\{\theta \mapsto \sum_i a_i \nabla\phi_i(\theta) : (a_1, \ldots, a_d) \in \mathbb{R}^d\right\} \tag{8}$$

where we write $\partial\phi(\theta)^\top = (\nabla\phi_1(\theta), \cdots, \nabla\phi_d(\theta)) \in \mathbb{R}^{D \times d}$, with $\nabla\phi_i \in \mathcal{C}^1(\Theta, \mathbb{R}^D)$.

The following results (which proofs can be found in Appendix C) establish that in some cases, the functions locally conserved for any data set are exactly the functions conserved through $W_\phi$.

**Lemma 2.13.** *Assume that the loss $(z, y) \mapsto \ell(z, y)$ is $\mathcal{C}^2$-differentiable with respect to $z \in \mathbb{R}^n$ and satisfies the condition:*

$$\operatorname*{span}_{y \in \mathcal{Y}}\{\nabla_z \ell(z, y)\} = \mathbb{R}^n, \forall z \in \mathbb{R}^n. \tag{9}$$

*Then for linear neural networks (resp. for two-layer ReLU networks) and all $\theta \in \Theta$ we have $W_{\phi(\theta)}^f = \mathbb{R}^d$, with the reparametrization $\phi$ from Example 2.10 and $\Theta := \mathbb{R}^D$ (resp. with $\phi$ from Example 2.11 and $\Theta$ consisting of all parameter $\theta$ of the network such that hidden neurons are associated to pairwise distinct "hyperplanes", cf Appendix C for details).*

Condition (9) holds for classical losses $\ell$ (e.g. quadratic/logistic losses), as shown in Lemma C.2 in Appendix C. Note that the additional hypothesis of pairwise distinct hyperplanes for the two-layer ReLU case is a generic hypothesis and is usual (see e.g. the notion of twin neurons in [27]). The tools from Appendix C extend Theorem 2.14 beyond (deep) linear and shallow ReLU networks. An open problem is whether the conclusions of Lemma 2.13 still hold for deep ReLU networks.

**Theorem 2.14.** *Under the same assumptions as in Lemma 2.13, we have that for linear neural networks, for all $\theta \in \Theta := \mathbb{R}^D$:*

$$W_\theta^g = W_\phi(\theta). \tag{10}$$

*The same result holds for two-layer ReLU networks with $\phi$ from Example 2.11 and $\Theta$ the (open) set of all parameters $\theta$ such that hidden neurons are associated to pairwise distinct "hyperplanes".*

This means as claimed that for linear and two-layer ReLU networks, being locally conserved on $\Theta$ for any data set exactly means being conserved on $\Theta$ through the finite-dimensional functional space $W_\phi \subseteq \mathcal{C}^1(\Theta, \mathbb{R}^D)$. This motivates the following definition

**Definition 2.15.** A real-valued function $h$ is a *conservation law of $\phi$* if it is conserved through $W_\phi$.

Proposition 2.5 yields the following intermediate result.

**Proposition 2.16.** $h \in \mathcal{C}^1(\Omega, \mathbb{R})$ *is a conservation law for $\phi$ if and only if*

$$\nabla h(\theta) \perp \nabla \phi_j(\theta), \ \forall\, \theta \in \Omega, \ \forall j \in \{1, \ldots, d\}.$$

Thanks to Theorem 2.14, the space $W_\phi$ defined in (8) introduces a much simpler proxy to express $W_\theta^g$ as a trace of a subset of $\mathcal{C}^1(\Theta, \mathbb{R}^D)$. Moreover, when $\phi$ is $\mathcal{C}^\infty$, $W_\phi$ is a *finite-dimensional* space of *infinitely smooth* functions on $\Theta$, and this will be crucial in Section 4.1 to provide a tractable scheme (i.e. operating in finite dimension) to compute the *maximum number of independent* conservation laws, using the Lie algebra computations that will be described in Section 3.

*Example* 2.17. Revisiting Example 2.8, the function to minimize is factorized by the reparametrization $\phi : (u \in \mathbb{R}, v \in \mathbb{R}) \mapsto uv \in \mathbb{R}$ with $\theta := (u, v)$. We saw that $h((u, v)) := u^2 - v^2$ is conserved: and indeed $\langle \nabla h(u, v), \nabla \phi(u, v) \rangle = 2uv - 2vu = 0, \forall (u, v)$.

In this simple example, the characterization of Proposition 2.16 gives a *constructive* way to find such a conserved function: we only need to find a function $h$ such that $\langle \nabla h(u, v), \nabla \phi(u, v) \rangle = \langle \nabla h(u, v), (v, u)^\top \rangle = 0$. The situation becomes more complex in higher dimensions, since one needs to understand the interplay between the different vector fields in $W_\phi$.

## 2.5 Constructibility of some conservation laws

Observe that in Example 2.17 both the reparametrization $\phi$ and the conservation law $h$ are polynomials, a property that surprisingly systematically holds in all examples of interest in the paper, making it possible to *algorithmically* construct some conservation laws as detailed now.

By Proposition 2.16, a function $h$ is a conservation law if it is in the kernel of the linear operator $h \in \mathcal{C}^1(\Omega, \mathbb{R}) \mapsto (\theta \in \Omega \mapsto (\langle \nabla h(\theta), \nabla \phi_i(\theta) \rangle)_{i=1,\cdots,d})$. Thus, one could look for conservation laws in a prescribed finite-dimensional space by projecting these equations in a basis (as in finite-element methods for PDEs). Choosing the finite-dimensional subspace could be generally tricky, but for the linear and ReLU cases all known conservation laws are actually polynomial "balancedness-type conditions" [1, 2, 9], see Section 4. In these cases, the vector fields in $W_\phi$ are also polynomials (because $\phi$ is polynomial, see Theorem C.4 and Theorem C.5 in Appendix C), hence $\theta \mapsto \langle \nabla h(\theta), \nabla \phi_i(\theta) \rangle$ is a polynomial too. This allows us to compute a basis of independent polynomial conservation laws of a given degree (to be freely chosen) for these cases, by simply focusing on the corresponding subspace of polynomials. We coded the resulting equations in SageMath, and we found back on selected examples (see Appendix J) all existing known conservation laws both for ReLU and linear networks. Open-source code is available at [18].

## 2.6 Independent conserved functions

Having an algorithm to build conservation laws is nice, yet how can we know if we have built "all" laws? This requires first defining a notion of a "maximal" set of functions, which would in some sense be independent. This does not correspond to linear independence of the functions themselves (for instance, if $h$ is a conservation law, then so is $h^k$ for each $k \in \mathbb{N}$ but this does not add any other constraint), but rather to pointwise linear independence of their gradients. This notion of independence is closely related to the notion of "functional independence" studied in [7, 20]. For instance, it is shown in [20] that smooth functionally dependent functions are characterized by having dependent gradients everywhere. This motivates the following definition.

**Definition 2.18.** A family of $N$ functions $(h_1, \cdots, h_N)$ conserved through $W \subset \mathcal{C}^1(\Omega, \mathbb{R}^D)$ is said to be *independent* if the vectors $(\nabla h_1(\theta), \cdots, \nabla h_N(\theta))$ are linearly independent for all $\theta \in \Omega$.

An immediate upper bound holds on the largest possible number $N$ of functionally independent functions $h_1, \ldots, h_N$ conserved through $W$: for $\theta \in \Omega \subseteq \mathbb{R}^D$, the space $\mathrm{span}\{\nabla h_1(\theta), \ldots, \nabla h_N(\theta)\} \subseteq \mathbb{R}^D$ is of dimension $N$ (by independence) and (by Proposition 2.5) orthogonal to $W(\theta)$. Thus, it is necessary to have $N \leq D - \dim W(\theta)$. As we will now see, this bound can be tight *under additional assumptions on $W$ related to Lie brackets* (corresponding to the so-called Frobenius theorem). This will in turn lead to a characterization of the maximum possible $N$.

# 3 Conservation Laws using Lie Algebra

The study of hyper-surfaces trapping the solution of ODEs is a recurring theme in control theory, since the existence of such surfaces is the basic obstruction of controllability of such systems [6]. The basic result to study these surfaces is the so-called Frobenius theorem from differential calculus (See Section 1.4 of [13] for a good reference for this theorem). It relates the existence of such surfaces, and their dimensions, to some differential condition involving so-called "Lie brackets" $[u, v]$ between pairs of vector fields (see Section 3.1 below for a more detailed exposition of this operation). However, in most cases of practical interest (such as for instance matrix factorization), the Frobenius theorem is not suitable for a direct application to the space $W_\phi$ because its Lie bracket condition is not satisfied. To identify the number of independent conservation laws, one needs to consider the algebraic closure of $W_\phi$ under Lie brackets. The fundamental object of interest is thus the Lie algebra generated by the Jacobian vector fields, that we recall next. While this is only defined for vector fields with stronger smoothness assumption, the only consequence is that $\phi$ is required to be infinitely smooth, unlike the loss $\ell(\cdot, y)$ and the model $g(\cdot, x)$ that can be less smooth. All concretes examples of $\phi$ in this paper are polynomial hence indeed infinitely smooth.

**Notations**  Given a vector subspace of infinitely smooth vector fields $W \subseteq \mathcal{X}(\Theta) := \mathcal{C}^\infty(\Theta, \mathbb{R}^D)$, where $\Theta$ is an open subset of $\mathbb{R}^D$, we recall (cf Proposition 2.5) that its trace at some $\theta$ is the subspace

$$W(\theta) := \mathrm{span}\{\chi(\theta) : \chi \in W\} \subseteq \mathbb{R}^D. \tag{11}$$

For each open subset $\Omega \subseteq \Theta$, we introduce the subspace of $\mathcal{X}(\Omega)$: $W_{|\Omega} := \{\chi_{|\Omega} : \chi \in W\}$.

## 3.1  Background on Lie algebra

A Lie algebra $A$ is a vector space endowed with a bilinear map $[\cdot, \cdot]$, called a Lie bracket, that verifies for all $X, Y, Z \in A$: $[X, X] = 0$ and the Jacobi identity: $[X, [Y, Z]] + [Y, [Z, X]] + [Z, [X, Y]] = 0$.

For the purpose of this article, the Lie algebra of interest is the set of infinitely smooth vector fields $\mathcal{X}(\Theta)$, endowed with the Lie bracket $[\cdot, \cdot]$ defined by

$$[\chi_1, \chi_2]: \quad \theta \in \Theta \mapsto [\chi_1, \chi_2](\theta) := \partial\chi_1(\theta)\chi_2(\theta) - \partial\chi_2(\theta)\chi_1(\theta), \tag{12}$$

with $\partial\chi(\theta) \in \mathbb{R}^{D \times D}$ the jacobian of $\chi$ at $\theta$. The space $\mathbb{R}^{n \times n}$ of matrices is also a Lie algebra endowed with the Lie bracket $[A, B] := AB - BA$. This can be seen as a special case of (12) in the case of *linear* vector fields, i.e. $\chi(\theta) = A\theta$.

**Generated Lie algebra**  Let $A$ be a Lie algebra and let $W \subset A$ be a vector subspace of $A$. There exists a smallest Lie algebra that contains $W$. It is denoted $\mathrm{Lie}(W)$ and called the generated Lie algebra of $W$. The following proposition [6, Definition 20] constructively characterizes $\mathrm{Lie}(W)$, where for vector subspaces $[W, W'] := \{[\chi_1, \chi_2] : \chi_1 \in W, \chi_2 \in W'\}$, and $W + W' = \{\chi_1 + \chi_2 : \chi_1 \in W, \chi_2 \in W'\}$.

**Proposition 3.1.** *Given any vector subspace $W \subseteq A$ we have $\mathrm{Lie}(W) = \bigcup_k W_k$ where:*
$$\begin{cases} W_0 & := W \\ W_k & := W_{k-1} + [W_0, W_{k-1}] \textit{ for } k \geq 1. \end{cases}$$

We will see in Section 3.2 that the number of conservation laws is characterized by the dimension of the trace $\mathrm{Lie}(W_\phi)(\theta)$ defined in (11). The following lemma (proved in Appendix D) gives a stopping criterion to algorithmically determine this dimension (see Section 3.3 for the algorithm).

**Lemma 3.2.** *Given $\theta \in \Theta$, if for a given $i$, $\dim W_{i+1}(\theta') = \dim W_i(\theta)$ for every $\theta'$ in a neighborhood of $\theta$, then there exists a neighborhood $\Omega$ of $\theta$ such that $W_k(\theta') = W_i(\theta')$ for all $\theta' \in \Omega$ and $k \geq i$, where the $V_i$ are defined by Proposition 3.1. Thus $\mathrm{Lie}(W)(\theta') = W_i(\theta')$ for all $\theta' \in \Omega$. In particular, the dimension of the trace of $\mathrm{Lie}(W)$ is locally constant and equal to the dimension of $W_i(\theta)$.*

## 3.2 Number of conservation laws

The following theorem uses the Lie algebra generated by $W_\phi$ to characterize the number of conservation laws. The proof of this result is based on two successive uses of the Frobenius theorem and can be found in Appendix E (where we also recall Frobenius theorem for the sake of completeness).

**Theorem 3.3.** *If* $\dim(\mathrm{Lie}(W_\phi)(\theta))$ *is locally constant then each* $\theta \in \Omega \subseteq \mathbb{R}^D$ *admits a neighborhood* $\Omega'$ *such that there are* $D - \dim(\mathrm{Lie}(W_\phi)(\theta))$ *(and no more) independent conserved functions through* $W_{\phi_{|\Omega'}}$, *i.e., there are* $D - \dim(\mathrm{Lie}(W_\phi)(\theta))$ *independent conservation laws of* $\phi$ *on* $\Omega'$.

*Remark* 3.4. The proof of the Frobenius theorem (and therefore of our generalization Theorem 3.3) is actually constructive. From a given $\phi$, conservation laws are obtained in the proof by integrating in time (*i.e.* solving an advection equation) the vector fields belonging to $W_\phi$. Unfortunately, this cannot be achieved in *closed form* in general, but in small dimensions, this could be carried out numerically (to compute approximate discretized laws on a grid or approximate them using parametric functions such as Fourier expansions or neural networks).

A fundamental aspect of Theorem 3.3 is to rely only on the *dimension of the trace* of the Lie algebra associated with the finite-dimensional vector space $W_\phi$. Yet, even if $W_\phi$ is finite-dimensional, it might be the case that $\mathrm{Lie}(W_\phi)$ itself remains infinite-dimensional. Nevertheless, what matters is not the dimension of $\mathrm{Lie}(W_\phi)$, but that of *its trace* $\mathrm{Lie}(W_\phi)(\theta)$, which is *always* finite (and potentially much smaller that $\dim \mathrm{Lie}(W_\phi)$ even when the latter is finite) and computationally tractable thanks to Lemma 3.2 as detailed in Section 3.3. In section 4.1 we work out the example of matrix factorization, a non-trivial case where the full Lie algebra $\mathrm{Lie}(W_\phi)$ itself remains finite-dimensional.

Theorem 3.3 requires that the dimension of the trace at $\theta$ of the Lie algebra is locally constant. This is a technical assumption, which typically holds outside a set of pathological points. A good example is once again matrix factorization, where we show in Section 4.1 that this condition holds generically.

## 3.3 Method and algorithm, with examples

Given a reparametrization $\phi$ for the architectures to train, to determine the number of independent conservation laws of $\phi$, we leverage the characterization 3.1 to algorithmically compute $\dim(\mathrm{Lie}(W_\phi)(\theta))$ using an iterative construction of bases for the subspaces $W_k$ starting from $W_0 := W_\phi$, and stopping as soon as the dimension stagnates thanks to Lemma 3.2. Our open-sourced code is available at [18] and uses SageMath. As we now show, this algorithmic principle allows to fully work out certain settings where the stopping criterion of Lemma 3.2 is reached at the first step ($i = 0$) or the second one ($i = 1$). Section 4.2 also discusses its numerical use for an empirical investigation of broader settings.

**Example where the iterations of Lemma 3.2 stop at the first step.** This corresponds to the case where $\mathrm{Lie} W_\phi(\theta) = W_1(\theta) = W_0(\theta) := W_\phi(\theta)$ on $\Omega$. This is the case if and only if $W_\phi$ satisfies that

$$[\chi_1, \chi_2](\theta) := \partial \chi_1(\theta) \chi_2(\theta) - \partial \chi_2(\theta) \chi_1(\theta) \in W_\phi(\theta), \quad \text{for all } \chi_1, \chi_2 \in W_\phi \text{ and all } \theta \in \Omega. \quad (13)$$

i.e., when Frobenius Theorem (see Theorem E.1 in Appendix E) applies directly. The first example is a follow-up to Example 2.11.

*Example* 3.5 (two-layer ReLU networks without bias). Consider $\theta = (U, V)$ with $U \in \mathbb{R}^{n \times r}, V \in \mathbb{R}^{m \times r}$, $n, m, r \geq 1$ (so that $D = (n + m)r$), and the reparametrization $\phi(\theta) := (u_i v_i^\top)_{i=1,\cdots,r} \in \mathbb{R}^{n \times m \times r}$, where $U = (u_1; \cdots; u_r)$ and $V = (v_1; \cdots; v_r)$. As detailed in Appendix F.1, since $\phi(\theta)$ is a collection of $r$ rank-one $n \times m$ matrices, $\dim(W_\phi(\theta)) = \mathtt{rank} \partial \phi(\theta) = (n + m - 1)r$ is constant on the domain $\Omega$ such that $u_i, v_j \neq 0$, and $W_\phi$ satisfies (13), hence by Theorem 3.3 each $\theta$ has a neighborhood $\Omega'$ such that there exists $r$ (and no more) independent conserved function through $W_{\phi_{|\Omega'}}$. The $r$ known conserved functions [9] given by $h_i : (U, V) \mapsto \|u_i\|^2 - \|v_i\|^2, i = 1, \cdots, r$, are independent, hence they are complete.

**Example where the iterations of Lemma 3.2 stop at the second step (but not the first one).** Our primary example is matrix factorization, as a follow-up to Example 2.10.

*Example* 3.6 (two-layer *linear* neural networks). With $\theta = (U, V)$, where $(U \in \mathbb{R}^{n \times r}, V \in \mathbb{R}^{m \times r})$ the reparameterization $\phi(\theta) := UV^\top \in \mathbb{R}^{n \times m}$ (here $d = nm$) factorizes the functions minimized during the training of linear two-layer neural networks (see Example 2.10). As shown in Appendix I, condition (13) is not satisfied when $r > 1$ and $\max(n, m) > 1$. Thus, the stopping criterion of Lemma 3.2 is not satisfied at the first step. However, as detailed in Proposition H.3 in Appendix H, $(W_\phi)_1 = (W_\phi)_2 = \mathrm{Lie}(W_\phi)$, hence the iterations of Lemma 3.2 stop at the second step.

We complete this example in the next section by showing (Corollary 4.4) that known conservation laws are indeed complete. Whether known conservation laws remain valid and/or *complete* in this settings and extended ones is further studied in Section 4 and Appendix F.

## 3.4 Application: recasting over-parameterized flows as low-dimensional Riemannian flows

As we now show, one striking application of Theorem 3.3 (in simple cases where $\dim(W_\phi(\theta)) = \dim(\text{Lie}W_\phi(\theta))$ is constant on $\Omega$, i.e., $\text{rank}(\partial\phi(\theta))$ is constant on $\Omega$ and $W_\phi$ satisfies (13)) is to fully rewrite the high-dimensional flow $\theta(t) \in \mathbb{R}^D$ as a low-dimensional flow on $z(t) \coloneqq \phi(\theta(t)) \in \mathbb{R}^d$, where this flow is associated with a Riemannian metric tensor $M$ that is induced by $\phi$ and depends on the initialization $\theta_{\text{init}}$. We insist on the fact that this is only possible in very specific cases, but this phenomenon is underlying many existing works that aim at writing in closed form the implicit bias associated with some training dynamics (see Section 1 for some relevant literature. Our analysis sheds some light on cases where this is possible, as shown in the next proposition.

**Proposition 3.7.** *Assume that $\text{rank}(\partial\phi(\theta))$ is constant on $\Omega$ and that $W_\phi$ satisfies (13). If $\theta(t) \in \mathbb{R}^D$ satisfies the ODE (2) where $\theta_{\text{init}} \in \Omega$, then there is $0 < T^\star_{\theta_{\text{init}}} \leq T_{\theta_{\text{init}}}$ such that $z(t) \coloneqq \phi(\theta(t)) \in \mathbb{R}^d$ satisfies the ODE*

$$\dot{z}(t) = -M(z(t), \theta_{\text{init}})\nabla f(z(t)) \quad \text{for all } t \in [0, T^\star_{\theta_{\text{init}}}), \text{ with } z(0) = \phi(\theta_{\text{init}}), \qquad (14)$$

*where $M(z(t), \theta_{\text{init}}) \in \mathbb{R}^{d \times d}$ is a symmetric positive semi-definite matrix.*

See Appendix G for a proof. Revisiting Example 3.5 leads to the following analytic example.

*Example* 3.8. Given the reparametrization $\phi : (u \in \mathbb{R}^*, v \in \mathbb{R}^d) \mapsto uv \in \mathbb{R}^d$, the variable $z \coloneqq uv$ satisfies (14) with $M(z, \theta_{\text{init}}) = \|z\|_\delta I_d + \|z\|_\delta^{-1} zz^\top$, with $\|z\|_\delta \coloneqq \delta + \sqrt{\delta^2 + \|z\|^2}$, $\delta \coloneqq 1/2(u_{\text{init}}^2 - \|v_{\text{init}}\|^2)$.

Another analytic example is discussed in Appendix G. In light of these results, an interesting perspective is to better understand the dependance of the Riemannian metric with respect to initialization, to possibly guide the choice of initialization for better convergence dynamics.

Note that the metric $M(z, \theta_{\text{init}})$ can have a kernel. Indeed, in practice, while $\phi$ is a function from $\mathbb{R}^D$ to $\mathbb{R}^d$, the dimensions often satisfy $\text{rank}\partial\phi(\theta) < \min(d, D)$, i.e., $\phi(\theta)$ lives in a manifold of lower dimension. The evolution (14) should then be understood as a flow on this manifold. The kernel of $M(z, \theta_{\text{init}})$ is orthogonal to the tangent space at $z$ of this manifold.

## 4 Conservation Laws for Linear and ReLU Neural Networks

To showcase the impact of our results, we show how they can be used to determine whether known conservation laws for linear (resp. ReLU) neural networks are complete, and to recover these laws *algorithmically* using reparametrizations $\phi$ adapted to these two settings. Concretely, we study the conservation laws for neural networks with $q$ layers, and either a linear or ReLU activation, with an emphasis on $q = 2$. We write $\theta = (U_1, \cdots, U_q)$ with $U_i \in \mathbb{R}^{n_{i-1} \times n_i}$ the weight matrices and we assume that $\theta$ satisfies the gradient flow (2). In the linear case the reparametrization is $\phi_{\text{Lin}}(\theta) \coloneqq U_1 \cdots U_q$. For ReLU networks, we use the (polynomial) reparametrization $\phi_{\text{ReLu}}$ of [27, Definition 6], which is defined for any (deep) feedforward ReLU network, with or without bias. In the simplified setting of networks without biases it reads explicitly as:

$$\phi_{\text{ReLu}}(U_1, \cdots, U_q) \coloneqq \left( U_1[:, j_1]U_2[j_1, j_2] \cdots U_{q-1}[j_{q-2}, j_{q-1}]U_q[j_{q-1}, :] \right)_{j_1, \cdots, j_{q-1}} \qquad (15)$$

with $U[i, j]$ the $(i, j)$-th entry of $U$. This covers $\phi(\theta) \coloneqq (u_j v_j^\top)_{j=1}^r \in \mathbb{R}^{n \times m \times r}$ from Example 2.11.

Some conservation laws are known for the linear case $\phi_{\text{Lin}}$ [1, 2] and for the ReLu case $\phi_{\text{ReLu}}$ [9].

**Proposition 4.1** ( [1, 2, 9] ). *If $\theta \coloneqq (U_1, \cdots, U_q)$ satisfies the gradient flow (2), then for each $i = 1, \cdots, q-1$ the function $\theta \mapsto U_i^\top U_i - U_{i+1} U_{i+1}^\top$ (resp. the function $\theta \mapsto \text{diag}\left(U_i^\top U_i - U_{i+1} U_{i+1}^\top\right)$) defines $n_i \times (n_i + 1)/2$ conservation laws for $\phi_{\text{Lin}}$ (resp. $n_i$ conservation laws for $\phi_{\text{ReLu}}$).*

Proposition 4.1 defines $\sum_{i=1}^{q-1} n_i \times (n_i + 1)/2$ conserved functions for the linear case. In general they are *not* independent, and we give below in Proposition 4.2, for the case of $q = 2$, the *exact*

number of independent conservation laws among these particular laws. Establishing whether there are other (previously unknown) conservation laws is an open problem for $q > 2$. We already answered negatively to this question in the two-layer ReLu case without bias (See Example 3.5). In the following Section (Corollary 4.4), we show the same result in the linear case $q = 2$. Numerical computations suggest this is still the case for deeper linear and ReLU networks as detailed in Section 4.2.

### 4.1 The matrix factorization case ($q = 2$)

To simplify the analysis when $q = 2$, we rewrite $\theta = (U, V)$ as a vertical matrix concatenation denoted $(U; V) \in \mathbb{R}^{(n+m) \times r}$, and $\phi(\theta) = \phi_{\texttt{Lin}}(\theta) = UV^\top \in \mathbb{R}^{n \times m}$.

**How many independent conserved functions are already known?**  The following proposition refines Proposition 4.1 for $q = 2$ by detailing how many *independent* conservation laws are already known. See Appendix H.1 for a proof.

**Proposition 4.2.** *Consider $\Psi : \theta = (U; V) \mapsto U^\top U - V^\top V \in \mathbb{R}^{r \times r}$ and assume that $(U; V)$ has full rank noted* `rk`. *Then the function $\Psi$ gives* `rk` $\cdot (2r + 1 - $ `rk`$)/2$ *independent conserved functions.*

**There exist no more independent conserved functions.**  We now come to the core of the analysis, which consists in actually computing $\text{Lie}(W_\phi)$ as well as its traces $\text{Lie}(W_\phi)(\theta)$ in the matrix factorization case. The crux of the analysis, which enables us to fully work out theoretically the case $q = 2$, is that $W_\phi$ is composed of *linear* vector fields (that are explicitly characterized in Proposition H.2 in Appendix H), the Lie bracket between two linear fields being itself linear and explicitly characterized with skew matrices, see Proposition H.3 in Appendix H. Eventually, what we need to compute is the dimension of the trace $\text{Lie}(W_\phi)(U, V)$ for any $(U, V)$. We prove the following in Appendix H.

**Proposition 4.3.** *If $(U; V) \in \mathbb{R}^{(n+m) \times r}$ has full rank noted* `rk`*, then:* $\dim(\text{Lie}(W_\phi)(U; V)) = (n + m)r - (2r + 1 - $ `rk`$)/2$.

With this explicit characterization of the trace of the generated Lie algebra and Proposition 4.2, we conclude that Proposition 4.1 has indeed exhausted the list of independent conservation laws.

**Corollary 4.4.** *If $(U; V)$ has full rank, then all conserved functions are given by $\Psi : (U, V) \mapsto U^\top U - V^\top V$. In particular, there exist no more* independent *conserved functions.*

### 4.2 Numerical guarantees in the general case

The expressions derived in the previous section are specific to the linear case $q = 2$. For deeper linear networks and for ReLU networks, the vector fields in $W_\phi$ are non-linear polynomials, and computing Lie brackets of such fields can increase the degree, which could potentially make the generated Lie algebra infinite-dimensional. One can however use Lemma 3.2 and stop as soon as $\dim\left((W_\phi)_k(\theta)\right)$ stagnates. Numerically comparing this dimension with the number $N$ of independent conserved functions known in the literature (predicted by Proposition 4.1) on a sample of depths/widths of small size, we empirically confirmed that there are no more conservation laws than the ones already known for deeper linear networks and for ReLU networks too (see Appendix J for details). Our code is open-sourced and is available at [18]. It is worth mentioning again that in all tested cases $\phi$ is polynomial, and there is a maximum set of conservation laws that are also polynomial, which are found algorithmically (as detailed in Section 2.5).

## Conclusion

In this article, we proposed a constructive program for determining the number of conservation laws. An important avenue for future work is the consideration of more general classes of architectures, such as deep convolutional networks, normalization, and attention layers. Note that while we focus in this article on gradient flows, our theory can be applied to any space of displacements in place of $W_\phi$. This could be used to study conservation laws for flows with higher order time derivatives, for instance gradient descent with momentum, by lifting the flow to a higher dimensional phase space. A limitation that warrants further study is that our theory is restricted to continuous time gradient flow. Gradient descent with finite step size, as opposed to continuous flows, disrupts exact conservation. The study of approximate conservation presents an interesting avenue for future work.

## Acknowledgement

The work of G. Peyré was supported by the European Research Council (ERC project NORIA) and the French government under management of Agence Nationale de la Recherche as part of the "Investissements d'avenir" program, reference ANR-19-P3IA-0001 (PRAIRIE 3IA Institute). The work of R. Gribonval was partially supported by the AllegroAssai ANR project ANR-19-CHIA-0009. We thank Thomas Bouchet for introducing us to SageMath, as well as Léo Grinsztajn for helpful feedbacks regarding the numerics. We thank Pierre Ablin and Raphaël Barboni for comments on a draft of this paper. We also thank the anonymous reviewers for their fruitful feedback.

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
