# A Proof of Proposition 2.5

Proposition 2.5 is a direct consequence of the following lemma (remember that $\nabla h(\theta) = [\partial h(\theta)]^\top$).

**Lemma A.1** (Smooth functions conserved through a given flow.). *Given $\chi \in \mathcal{C}^1(\Omega, \mathbb{R}^D)$, a function $h \in \mathcal{C}^1(\Omega, \mathbb{R})$ is conserved through the flow induced by $\chi$ if and only if $\partial h(\theta)\chi(\theta) = 0$ for all $\theta \in \Omega$.*

*Proof.* Assume that $\partial h(\theta)\chi(\theta) = 0$ for all $\theta \in \Omega$. Then for all $\theta_{\text{init}} \in \Omega$ and for all $t \in (0, T_{\theta_{\text{init}}})$ :

$$\frac{\mathrm{d}}{\mathrm{d}t} h(\theta(t, \theta_{\text{init}})) = \partial h(\theta(t, \theta_{\text{init}}))\dot{\theta}(t, \theta_{\text{init}}) = \partial h(\theta(t, \theta_{\text{init}}))\chi(\theta(t, \theta_{\text{init}})) = 0.$$

Thus: $h(\theta(t, \theta_{\text{init}})) = h(\theta_{\text{init}})$, *i.e.*, $h$ is conserved through $\chi$. Conversely, assume that there exists $\theta_0 \in \Omega$ such that $\partial h(\theta_0)\chi(\theta_0) \neq 0$. Then by continuity of $\theta \in \Omega \mapsto \partial h(\theta)\chi(\theta)$, there exists $r > 0$ such that $\partial h(\theta)\chi(\theta) \neq 0$ on $B(\theta_0, r)$. With $\theta_{\text{init}} = \theta_0$ by continuity of $t \mapsto \theta(t, \theta_{\text{init}})$, there exists $\varepsilon > 0$, such that for all $t < \varepsilon$, $\theta(t, \theta_{\text{init}}) \in B(\theta_0, r)$. Then for all $t \in (0, \varepsilon)$: $\frac{\mathrm{d}}{\mathrm{d}t} h(\theta(t, \theta_{\text{init}})) = \partial h(\theta(t, \theta_{\text{init}}))\chi(\theta(t, \theta_{\text{init}})) \neq 0$, hence $h$ is not conserved through the flow induced by $\chi$. $\square$

# B Proof of Proposition 2.7

**Proposition B.1.** *Assume that for each $y \in \mathcal{Y}$ the loss $\ell(z, y)$ is $\mathcal{C}^2$-differentiable with respect to $z \in \mathbb{R}^n$. For each $\theta \in \Theta$ we have:*

$$W_\theta^g = \operatorname*{span}_{(x,y) \in \mathcal{X}_\theta \times \mathcal{Y}} \{[\partial_\theta g(\theta, x)]^\top \nabla_z \ell(g(\theta, x), y)\}$$

*where $\mathcal{X}_\theta$ is the set of data points $x$ such that $g(\cdot, x)$ is $\mathcal{C}^2$-differentiable in the neighborhood of $\theta$.*

*Proof.* Let us first show the direct inclusion. Let $\Omega \subseteq \Theta$ be a neighborhood of $\theta$ and let $\chi \in W_\Omega^g$. Let us show that $\chi(\theta) \in \operatorname*{span}_{(x,y) \in \mathcal{X}_\theta \times \mathcal{Y}} \{[\partial_\theta g(\theta, x)]^\top \nabla_z \ell(g(\theta, x), y)\}$. As $\chi \in W_\Omega^g$, there exist $X = (x_i)_i, Y = (y_i)_i$ such that $\forall i \; g(\cdot, x_i) \in \mathcal{C}^2(\Omega, \mathbb{R})$ (and thus $x_i \in \mathcal{X}_\theta$) and $\chi(\cdot) = \nabla \mathcal{E}_{X,Y}(\cdot) \in \mathcal{C}^1(\Omega, \mathbb{R}^D)$ (cf (3)). Moreover, for each $\theta' \in \Omega$, by chain rules and (1), we have:

$$\nabla \mathcal{E}_{X,Y}(\theta') = \sum_i [\partial_\theta g(\theta', x_i)]^\top \nabla_z \ell(g(\theta', x_i), y_i),$$

where $x_i \in \mathcal{X}_\theta$. Thus $\chi(\theta) \in \operatorname*{span}_{(x,y) \in \mathcal{X}_\theta \times \mathcal{Y}} \{[\partial_\theta g(\theta, x)]^\top \nabla_z \ell(g(\theta, x), y)\}$. This leads to the direct inclusion.

Now let us show the converse inclusion. Let $(x, y) \in \mathcal{X}_\theta \times \mathcal{Y}$. Let us show that $[\partial_\theta g(\theta, x)]^\top \nabla_z \ell(g(\theta, x), y) \in W_\theta^g$. By definition of $\mathcal{X}_\theta$, there exists a neighborhood $\Omega$ of $\theta$ such that $g(\cdot, x) \in \mathcal{C}^2(\Omega, \mathbb{R}^D)$. By taking $X = x$ and $Y = y$ (*i.e.* a data set of one feature and one target), one has still by chain rules $\nabla \mathcal{E}_{X,Y}(\cdot) = [\partial_\theta g(\cdot, x)]^\top \nabla_z \ell(g(\cdot, x), y) \in W_\Omega^g$. Finally by definition (4) of the trace and by (5), $[\partial_\theta g(\theta, x)]^\top \nabla_z \ell(g(\theta, x), y) = \nabla \mathcal{E}_{X,Y}(\theta) \in W_\Omega^g(\theta) \subseteq W_\theta^g$ as $\Omega$ is a neighborhood of $\theta$. $\square$

# C Proof of Lemma 2.13 and Theorem 2.14

We recall (cf Example 2.10 and Example 2.11) that linear and 2-layer ReLU neural networks satisfy Assumption 2.9, which we recall reads as:

**Assumption 2.9** (Local reparameterization) For each parameter $\theta_0 \in \mathbb{R}^D$, for each $x \in \mathcal{X}_{\theta_0}$, there is a neighborhood $\Omega$ of $\theta_0$ and a function $f(\cdot, x) \in \mathcal{C}^2(\phi(\Omega), \mathbb{R}^n)$ such that

$$\forall \theta \in \Omega, \quad g(\theta, x) = f(\phi(\theta), x), \tag{16}$$

where we also recall that

$$\mathcal{X}_{\theta_0} := \{x \in \mathcal{X} : \theta \mapsto g(\theta, x) \text{ is } \mathcal{C}^2 \text{ in the neighborhood of } \theta_0\}. \tag{17}$$

A common assumption to Lemma 2.13 and Theorem 2.14 is that the loss $\ell(z, y)$ is such that $\ell(\cdot, y)$ is $\mathcal{C}^2$-differentiable for all $y$, hence by Proposition 2.7 and Proposition 2.12 we have

$$W_\theta^g = \operatorname*{span}_{(x,y)\in\mathcal{X}_\theta\times\mathcal{Y}} \{[\partial_\theta g(\theta, x)]^\top \nabla_z \ell(g(\theta, x), y)\} \quad \text{and} \quad W_\theta^g = \partial\phi(\theta)^\top W_{\phi(\theta)}^f$$

where

$$W_{\phi(\theta)}^f := \operatorname*{span}_{(x,y)\in\mathcal{X}_\theta\times\mathcal{Y}} \{\partial f^x(\phi(\theta))^\top \nabla_z \ell(g(\theta, x), y)\} \text{ and } f^x(\cdot) := f(\cdot, x).$$

**Consequence of the assumption (9).** To proceed further we will rely on the following lemma that shows a direct consequence of (9) (in addition to Assumption 2.9 on the model $g(\theta, \cdot)$).

**Lemma C.1.** *Under Assumption 2.9, considering a loss $\ell(z, y)$ such that $\ell(\cdot, y)$ is $\mathcal{C}^2$-differentiable for all $y$. Denote $f^x(\cdot) := f(\cdot, x)$. If the loss satisfies (9), i.e.*

$$\operatorname*{span}_{y\in\mathcal{Y}}\{\nabla_z \ell(z, y)\} = \mathbb{R}^n, \forall z \in \mathbb{R}^n,$$

*then for all $\theta \in \mathbb{R}^D$,*

$$W_{\phi(\theta)}^f = \operatorname*{span}_{(x,w)\in\mathcal{X}_\theta\times\mathbb{R}^n} \{\partial f^x(\phi(\theta))^\top w\} \tag{18}$$

*Proof.* For $\theta \in \mathbb{R}^D$, we have

$$
\begin{aligned}
W_{\phi(\theta)}^f &= \operatorname*{span}_{(x,y)\in\mathcal{X}_\theta\times\mathcal{Y}} \{\partial f^x(\phi(\theta))^\top \nabla_z \ell(g(\theta, x), y)\} \\
&= \operatorname*{span}_{x\in\mathcal{X}_\theta} \{\partial f^x(\phi(\theta))^\top \operatorname*{span}_{y\in\mathcal{Y}}\{\nabla_z \ell(g(\theta, x), y)\}\} \\
&\overset{(9)}{=} \operatorname*{span}_{x\in\mathcal{X}_\theta} \{\partial f^x(\phi(\theta))^\top \mathbb{R}^n\} \\
&= \operatorname*{span}_{(x,w)\in\mathcal{X}_\theta\times\mathbb{R}^n} \{\partial f^x(\phi(\theta))^\top w\}. \qquad \square
\end{aligned}
$$

**Verification of (9) for standard ML losses.** Before proceeding to the proof of Lemma 2.13 and Theorem 2.14, let us show that (9) holds for standard ML losses.

**Lemma C.2.** *The mean-squared error loss $(z, y) \mapsto \ell_2(z, y) := \|y - z\|^2$ and the logistic loss $(z \in \mathbb{R}, y \in \{-1, 1\}) \mapsto \ell_{\texttt{logis}}(z, y) := \log(1 + \exp(-zy))$ satisfy condition (9).*

*Proof.* To show that $\ell_2$ satisfies (9) we observe that, with $e_i$ the $i$-th canonical vector, we have

$$\mathbb{R}^n = \operatorname{span}\{e_i : 1 \le i \le n\} = \operatorname*{span}_{y\in\{z-e_i/2\}_{i=1}^n} 2(z - y) \subseteq \operatorname*{span}_{y\in\mathbb{R}^n} 2(z - y) = \operatorname*{span}_{y\in\mathbb{R}^n} \nabla_z \ell_2(z, y) \subseteq \mathbb{R}^n.$$

For the logistic loss, $\nabla_z \ell_{\texttt{logis}}(z, y) = \frac{-y\exp(-zy)}{1+\exp(-zy)} \ne 0$ hence $\operatorname{span}_y \nabla_z \ell_{\texttt{logis}}(z, y) = \mathbb{R}$. $\square$

*Remark* C.3. In the case of the cross-entropy loss $(z \in \mathbb{R}^n, y \in \{1, \cdots, n\}) \mapsto \ell_{\texttt{cross}}(z, y) := -z_y + \log\left(\sum_{i=1}^n \exp z_i\right)$, $\ell_{\texttt{cross}}$ *does not* satisfy (9) as $\nabla_z \ell_{\texttt{cross}}(z, y) = -e_y + \begin{pmatrix} \exp(z_1)/(\sum_i \exp z_i) \\ \cdots \\ \exp(z_n)/(\sum_i \exp z_i) \end{pmatrix}$ satisfies for all $z \in \mathbb{R}^n$:

$$\operatorname{span}_y \nabla_z \ell_{\texttt{cross}}(z, y) = \{w := (w_1, \cdots, w_n) \in \mathbb{R}^n : \sum w_i = 0\} =: L_{\texttt{cross}}.$$

An interesting challenge is to investigate variants of Lemma 2.13 under weaker assumptions that would cover the cross-entropy loss.

**The case of linear neural networks of any depth.** Let us first prove Lemma 2.13 and Theorem 2.14 for the case of linear neural networks.

**Theorem C.4** (linear networks)**.** *Consider a linear network parameterized by $q$ matrices, $\theta = (U_1, \ldots, U_q)$ and defined via $g(\theta, x) := U_1 \ldots U_q x$. With $\phi(\theta) := U_1 \ldots U_q \in \mathbb{R}^{n\times m}$ (identified with $\mathbb{R}^d$ with $d = nm$), and for any loss $\ell$ satisfying (9), we have for all $\theta \in \mathbb{R}^D$, $W_{\phi(\theta)}^f = \mathbb{R}^d$ and $W_\theta^g = \operatorname{range}(\partial\phi(\theta)^\top)$.*

*Proof.* Let $\theta \in \Theta = \mathbb{R}^D$. As we can factorize the model (cf Example 2.10) by $g(\cdot, x) = \phi(\cdot)x =: f^x(\phi(\cdot)) \in \mathbb{R}^n$ for $x \in \mathcal{X}_\theta = \mathbb{R}^m$. Thus, by using Lemma C.1: $W^f_{\phi(\theta)} = \mathrm{span}_{x \in \mathcal{X}_\theta, w \in \mathbb{R}^n}\{[\partial f^x(\phi(\theta))]^\top w\} = \mathrm{span}_{x \in \mathbb{R}^m, w \in \mathbb{R}^n}\{wx^\top\} = \mathbb{R}^d$. Finally by Proposition 2.12: $W^g_\theta = \partial\phi(\theta)^\top W^f_{\phi(\theta)} = \mathrm{range}(\partial\phi(\theta)^\top)$. $\qquad\square$

**The case of two-layer ReLU networks.** In the case of two-layer ReLU networks with $r$ neurons, one can write $\theta = (U, V, b, c) \in \mathbb{R}^{n \times r} \times \mathbb{R}^{m \times r} \times \mathbb{R}^r \times \mathbb{R}^n$, and denote $u_j$ (resp. $v_j$, $b_j$) the columns of $U$ (resp. columns of $V$, entries of $b$), so that $g_\theta(x) = \sum_{j=1}^r u_j \sigma(v_j^\top x + b_j) + c$. The set $\mathcal{X}_\theta$ (defined in (17)) is simply the complement in the input domain $\mathbb{R}^m$ of the union of the hyperplanes

$$\mathcal{H}_j := \{x \in \mathbb{R}^m : v_j^\top x + b_j = 0\}. \tag{19}$$

**Theorem C.5** (two-layer ReLU networks). *Consider a loss $\ell(z, y)$ satisfying (9) and such that $\ell(\cdot, y)$ is $\mathcal{C}^2$-differentiable for all $y$. On a two-layer ReLU network architecture, let $\theta$ be a parameter such that all hyperplanes $\mathcal{H}_j$ defined in (19) are pairwise distinct. Then, with $\phi_{\mathtt{ReLU}}$ the reparameterization of Example 2.11, we have: $W^f_{\phi_{\mathtt{ReLU}}(\theta)} = \mathbb{R}^d$ and $W^g_\theta = \mathrm{range}(\partial\phi_{\mathtt{ReLU}}(\theta)^\top)$.*

*Proof.* Let $\theta$ be a parameter such that all hyperplanes $\mathcal{H}_j$ defined in (19) are pairwise distinct. Since the loss $\ell$ satisfies (9), by Lemma C.1, we only need to show that:

$$\mathrm{span}_{(x,w) \in \mathcal{X}_\theta \times \mathbb{R}^n} \{\partial f^x(\phi(\theta))^\top w\} = \mathbb{R}^d.$$

For convenience we will use the shorthand $C_{\theta, x}$ for the Jacobian matrix $\partial f^x(\phi(\theta))$.

*1st case: We consider first the case without bias ($b_j, c = 0$).* In that case, by Example 2.11 we have $\phi(\theta) := (u_j v_j^\top)_{j=1}^r$ where we write: $\theta = (U, V) \in \mathbb{R}^{n \times r} \times \mathbb{R}^{m \times r}$, and denote $u_j$ (resp. $v_j$) the columns of $U$ (resp. columns of $V$). Here $d = rnm$ and it can be checked (see Example 2.11) that

$$C_{\theta, x}^\top := \begin{pmatrix} \varepsilon_1(x, \theta) A(x) \\ \cdots \\ \varepsilon_r(x, \theta) A(x) \end{pmatrix} \in \mathbb{R}^{(rnm) \times n}$$

where:

$$A : x \in \mathbb{R}^m \mapsto A(x) := \begin{pmatrix} x & 0 & \cdots & 0 \\ 0 & x & \cdots & 0 \\ \cdots & \cdots & \cdots & \cdots \\ 0 & \cdots & 0 & x \end{pmatrix} \in \mathbb{R}^{(nm) \times n},$$

and where $\varepsilon_i(x, \theta) = \mathbb{1}(v_i^\top x > 0)$. For $j = 1, \cdots, r$ we denote:

$$\mathcal{A}_j^+ := \{x \in \mathbb{R}^m : v_j^\top x > 0\}, \quad \text{and} \quad \mathcal{A}_j^- := \{x \in \mathbb{R}^m : v_j^\top x < 0\}.$$

The open Euclidean ball of radius $r > 0$ centered at $c \in \mathbb{R}^m$ is denoted $B(c, r)$.

Consider a hidden neuron $i \in \{1, \cdots, r\}$ and denote $\mathcal{H}_i' := \mathcal{H}_i - \left(\bigcup_{j \neq i} \mathcal{H}_j\right)$. Since the hyperplanes are pairwise distinct, $\mathcal{H}_i' \neq \emptyset$ so we can consider an arbitrary $x' \in \mathcal{H}_i'$. Given any $\eta > 0$, by continuity of $x \in \mathbb{R}^m \mapsto (v_1^\top x, \cdots, v_r^\top x) \in \mathbb{R}^r$, there exists $x_\eta^+ \in B(x', \eta) \cap \mathcal{A}_i^+$ and $x_\eta^- \in B(x', \eta) \cap \mathcal{A}_i^-$ such that for all $j \neq i$, $\mathrm{sign}(v_j^\top x_\eta^\pm) = \mathrm{sign}(v_j^\top x')$. It follows that $x_\eta^\pm \in \mathcal{X}_\theta$ (remember that $\mathcal{X}_\theta$ is the complement of $\cup_j \mathcal{H}_j$). As a consequence:

$$\begin{pmatrix} 0 \\ \cdots \\ 0 \\ A(x') \\ 0 \\ \cdots \\ 0 \end{pmatrix} = \lim_{\eta \to 0} \left(C_{\theta, x_\eta^+}^\top - C_{\theta, x_\eta^-}^\top\right) \in \overline{\mathrm{span}\{C_{\theta, x}^\top\}}_{x \in \mathcal{X}_\theta} = \mathrm{span}\{C_{\theta, x}^\top\}_{x \in \mathcal{X}_\theta},$$

where the nonzero line in the left-hand-side is the $i$-th, and we used that every finite-dimensional space is closed.

Moreover still by continuity of $x \in \mathbb{R}^m \mapsto (v_1^\top x, \cdots, v_r^\top x) \in \mathbb{R}^r$, there exists $\gamma > 0$, such that for $k = \{-2, -1, 1, 2\}$, the vectors defined as:

$$x_k := x' + \gamma k v_i,$$

satisfy for all $j \neq i$, $\mathrm{sign}(v_j^\top x_k) = \mathrm{sign}(v_j^\top x')$ and $v_i^\top x_k \neq 0$, so that $x_k \in \mathcal{X}_\theta$ and we similarly obtain

$$\begin{pmatrix} 0 \\ \cdots \\ 0 \\ \gamma A(v_i) \\ 0 \\ \cdots \\ 0 \end{pmatrix} = C_{\theta,x_2}^\top - C_{\theta,x_1}^\top - \left( C_{\theta,x_{-1}}^\top - C_{\theta,x_{-2}}^\top \right) \in \underset{x \in \mathcal{X}_\theta}{\mathrm{span}}\{C_{\theta,x}^\top\}.$$

As this holds for every $x' \in \mathcal{H}_i'$, and since $\mathrm{span}\{v_i, \mathcal{H}_i'\} = \mathbb{R}^m$, we deduce that for any $x \in \mathbb{R}^m$

$$\begin{pmatrix} 0 \\ \cdots \\ 0 \\ A(x) \\ 0 \\ \cdots \\ 0 \end{pmatrix} \in \underset{x \in \mathcal{X}_\theta}{\mathrm{span}}\{C_{\theta,x}^\top\}.$$

As this holds for every hidden neuron $i = 1, \cdots, r$ it follows that for every $x^1, \cdots, x^r \in \mathbb{R}^m$

$$\begin{pmatrix} A(x^1) \\ \cdots \\ A(x^r) \end{pmatrix} \in \underset{x \in \mathcal{X}_\theta}{\mathrm{span}}\{C_{\theta,x}^\top\}.$$

Moreover, by definition of $A(\cdot)$, for each $x \in \mathbb{R}^m$ and each $w = (w_1, \cdots, w_n) \in \mathbb{R}^n$, we have

$$A(x)w = \begin{pmatrix} w_1 x \\ \cdots \\ w_n x \end{pmatrix} \in \mathbb{R}^{nm}.$$

Identifying $\mathbb{R}^{nm}$ with $\mathbb{R}^{m \times n}$ and the above expression with $xw^\top$, we deduce that

$$\underset{x \in \mathbb{R}^m, w \in \mathbb{R}^n}{\mathrm{span}} A(x)w = \mathbb{R}^{nm}$$

and we let the reader check that this implies

$$\underset{x^1, \cdots, x^r \in \mathbb{R}^m, w \in \mathbb{R}^n}{\mathrm{span}} \begin{pmatrix} A(x^1) \\ \cdots \\ A(x^r) \end{pmatrix} w = \underset{x^1, \cdots, x^r \in \mathbb{R}^m, w \in \mathbb{R}^n}{\mathrm{span}} \begin{pmatrix} A(x^1)w \\ \cdots \\ A(x^r)w \end{pmatrix} = \mathbb{R}^{rnm}.$$

Thus, as claimed, we have

$$\underset{x \in \mathcal{X}_\theta, w \in \mathbb{R}^n}{\mathrm{span}} \{C_{\theta,x}^\top w\} = \mathbb{R}^{rnm} = \mathbb{R}^d.$$

*2d case: General case with biases.* The parameter is $\theta = (U, V, b, c) \in \mathbb{R}^{n \times r} \times \mathbb{R}^{m \times r} \times \mathbb{R}^r \times \mathbb{R}^n$ with $b = (b_i)_{i=1}^r$, where $b_i \in \mathbb{R}$ the bias of the $i$-th hidden neuron, and $c$ the output bias.

In that case, $d = rn(m + 1)$ and one can check that the conditions of Assumption 2.9 hold with $\phi_{\mathrm{ReLU}}(\theta) := ((u_i v_i^\top, u_i b_i)_{i=1}^r, c)$ and $f^x(\phi) := C_{\theta,x}\phi$ where $C_{\theta,x}$ is expressed as:

$$C_{\theta,x}^\top := \begin{pmatrix} \varepsilon_1(x, \theta)A'(x) \\ \cdots \\ \varepsilon_r(x, \theta)A'(x) \\ I_n \end{pmatrix} \in \mathbb{R}^{(rn(m+1)+n) \times n}$$

where, denoting $\bar{x} = (x^\top, 1)^\top \in \mathbb{R}^{m+1}$, we defined

$$A' : x \in \mathbb{R}^m \mapsto A'(x) := \begin{pmatrix} \bar{x} & 0 & \cdots & 0 \\ 0 & \bar{x} & \cdots & 0 \\ \cdots & \cdots & \cdots & \cdots \\ 0 & \cdots & 0 & \bar{x} \end{pmatrix} \in \mathbb{R}^{n(m+1) \times n},$$

and $\varepsilon_i(x, \theta) := \mathbb{1}(v_i^\top x + b_i > 0)$.

Using the sets

$$\mathcal{A}_j^+ := \{x \in \mathbb{R}^m : v_j^\top x + b_j > 0\}, \quad \text{and} \quad \mathcal{A}_j^- := \{x \in \mathbb{R}^m : v_j^\top x + b_j < 0\},$$

a reasoning analog to the case without bias allows to show that for each $i = 1, \cdots, r$:

$$\operatorname*{span}_{x \in \mathbb{R}^m} \begin{pmatrix} 0 \\ \cdots \\ 0 \\ A'(x) \\ 0 \\ \cdots \\ 0 \end{pmatrix} \in \operatorname*{span}_{x \in \mathcal{X}_\theta}\{C_{\theta,x}^\top\}$$

so that, again, for every $x^1, \ldots, x^r \in \mathbb{R}^m$ we have

$$\begin{pmatrix} A'(x^1) \\ \cdots \\ A'(x^r) \\ 0 \end{pmatrix} \in \operatorname*{span}_{x \in \mathcal{X}_\theta}\{C_{\theta,x}^\top\}.$$

As $\begin{pmatrix} 0 \\ \cdots \\ 0 \\ I_n \end{pmatrix} \in \operatorname*{span}_{x \in \mathcal{X}_\theta}\{C_{\theta,x}^\top\}$ too, we obtain that $\begin{pmatrix} A'(x^1) \\ \cdots \\ A'(x^r) \\ I_n \end{pmatrix} \in \operatorname*{span}_{x \in \mathcal{X}_\theta}\{C_{\theta,x}^\top\}.$

Now, for each $x \in \mathbb{R}^m$ and $w = (w_1, \cdots, w_n) \in \mathbb{R}^n$, we have

$$A'(x)w = \begin{pmatrix} w_1 \bar{x} \\ \cdots \\ w_n \bar{x} \end{pmatrix} = \begin{pmatrix} w_1 x \\ w_1 \\ \cdots \\ w_n x \\ w_n \end{pmatrix} \in \mathbb{R}^{n(m+1)}.$$

Again, identifying the above expression with $w(x^\top, 1) \in \mathbb{R}^{n \times (m+1)}$ it is not difficult to check that

$$\operatorname*{span}_{x \in \mathbb{R}^m, w \in \mathbb{R}^n} A'(x)w = \mathbb{R}^{n(m+1)},$$

and we conclude as before.

In both cases we established that $W_{\phi_{\mathtt{ReLU}}}^f(\theta) = \mathbb{R}^d$. Finally by Proposition 2.12 we obtain $W_\theta^g = \partial\phi_{\mathtt{ReLU}}(\theta)^\top W_{\phi_{\mathtt{ReLU}}(\theta)}^f = \operatorname{range}(\partial\phi_{\mathtt{ReLU}}(\theta)^\top)$. $\qquad\square$

Combining Theorem C.4 and Theorem C.5 establishes Lemma 2.13 Theorem 2.14 as claimed. One can envision extensions of these results to deeper ReLU networks, using notations and concepts from [27] that generalize observations from Example 2.11 to deep ReLU networks with biases. Given a feedforward network architecture of arbitrary depth, denote $\theta$ the collection of all parameters (weights and biases) of a ReLU network on this architecture, and consider $\theta \mapsto \phi_{\mathtt{ReLU}}(\theta)$ the rescaling-invariant polynomial function of [27, Definition 6] and $C_{\theta,x}$, the matrices of [27, Corollary 3] such that the output of the network with parameters $\theta$, when fed with an input vector $x \in \mathbb{R}^m$, can be written $g(\theta, x) = C_{\theta,x}\phi(\theta)$. From its definition in [27, Corollary 3], given $x$, the matrix $C_{\theta,x}$ only depends on $\theta$ via the so-called *activation status* of the neurons in the network (cf [27, Section 4.1]).

# D Proof of Lemma 3.2

**Lemma D.1.** *Given $\theta \in \Theta$, if for a given $i$, $\dim(W_{i+1}(\theta')) = \dim(W_i(\theta))$ for every $\theta'$ in a neighborhood of $\theta$, then for all $k \geq i$, we have $W_k(\theta') = W_i(\theta')$ for all $\theta'$ in a neighborhood $\Omega$ of $\theta$, where the $W_i$ are defined by Proposition 3.1. Thus $\mathrm{Lie}(W)(\theta') = W_i(\theta')$ for all $\theta' \in \Omega$. In particular, the dimension of the trace of $\mathrm{Lie}(W)$ is locally constant and equal to the dimension of $W_i(\theta)$.*

*Proof.* The result is obvious for $k = i$. The proof is by induction on $k$ starting from $k = i + 1$. We denote $m := \dim(W_i(\theta))$.

*1st step: Initialization $k = i + 1$.* By definition of the spaces $W_i$ (cf Proposition 3.1) we have $W_i \subset W_{i+1}$ hence $W_i(\theta) \subseteq W_{i+1}(\theta)$. Since $\dim(W_{i+1}(\theta)) = \dim(W_i(\theta)) = m$, it follows that there exists $\chi_1, \cdots, \chi_m \in W_i$ such that $\mathrm{span}_j \chi_j(\theta) = W_i(\theta) = W_{i+1}(\theta)$ (hence the $m$

vectors $(\chi_1(\theta), \cdots, \chi_m(\theta))$ are linearly independent). Since each $\chi_j$ is smooth, it follows that $(\chi_1(\theta'), \cdots \chi_m(\theta'))$ remain linearly independent on some neighborhood $\Omega$ of $\theta$, which we assume to be small enough to ensure $\dim W_{i+1}(\theta') = m$ for all $\theta' \in \Omega$. As $\chi_j \in W_i \subset W_{i+1}$, we obtain that for each $\theta' \in \Omega$, the family $\{\chi_j(\theta')\}_{j=1}^m$ is a basis of the $m$-dimensional subspace $W_{i+1}(\theta')$, hence:

$$W_i(\theta') \subset W_{i+1}(\theta') = \mathrm{span}_j \chi_j(\theta') \subset W_i(\theta'), \quad \forall \theta' \in \Omega \tag{20}$$

*2nd step: Induction.* We assume $W_k(\theta') = W_i(\theta')$ on $\Omega$. Let us show that $W_{k+1}(\theta') = W_i(\theta')$ on $\Omega$. Since $W_{k+1} := W_k + [W_0, W_k]$ it is enough to show that $[W_0, W_k](\theta') \subseteq W_i(\theta')$ on $\Omega$. For this, considering two vector fields, $f \in W_0$ and $\chi \in W_k$, we will show that $[f, \chi](\theta') \in W_{i+1}(\theta')$ for each $\theta' \in \Omega$. In light of (20), this will allow us to conclude.

Indeed, from the induction hypothesis we know that $W_k(\theta') = \mathrm{span}_j \chi_j(\theta') = W_i(\theta')$ on $\Omega$, hence for each $\theta' \in \Omega$ there are coefficients $a_j(\theta')$ such that $\chi(\theta') = \sum_{j=1}^m a_j(\theta') \chi_j(\theta')$. Standard linear algebra shows that these coefficients depend smoothly on $\chi(\theta')$ and $\chi_j(\theta')$, which are smooth functions of $\theta'$, hence the functions $a_j(\cdot)$ are smooth. By linearity of the Lie bracket and of $W_{i+1}(\theta')$ it is enough to show that $[f, a_j \chi_j](\theta') \in W_{i+1}(\theta')$ on $\Omega$ for each $j$. Standard calculus yields

$$[f, a_j\chi_j] = (\partial f)(a_j\chi_j) - \underbrace{\partial(a_j\chi_j)}_{=\chi_j \partial a_j + a_j \partial \chi_j} f = a_j[(\partial f)\chi_j - (\partial \chi_j)f] - \chi_j(\partial a_j)f$$
$$= a_j[f, \chi_j] - [(\partial a_j)f]\chi_j$$

since $(\partial a_j)f$ is scalar-valued (consider the corresponding dimensions). Since $f \in W_0$ and $\chi_j \in W_i$, by definition of $W_{i+1}$ (cf Proposition 3.1) we have $[f, \chi_j], \chi_j \in W_{i+1}$ hence by linearity we conclude that $[f, a_j\chi_j](\theta') \in W_{i+1}(\theta')$. As this holds for all $j$, we obtain $[f, \chi](\theta') \in W_{i+1}(\theta')$. As this is valid for any $f \in W_0$, $\chi \in W_k$ this establishes $[W_0, W_k](\theta') \subseteq W_{i+1}(\theta') \overset{(20)}{\subseteq} W_i(\theta')$ and we conclude as claimed that $W_i(\theta') \subseteq W_{k+1}(\theta') = W_k(\theta') + [W_0, W_k](\theta') \subseteq W_i(\theta')$ on $\Omega$. $\qquad \square$

# E Proof of Theorem 3.3

We recall first the fundamental result of Frobenius using our notations (See Section 1.4 of [13]). When we refer to a "non-singular distribution", it implies that the dimension of the associated trace remains constant (refer to the definition of "non-singular" on page 15 of [13]). Being "involutively consistent" directly relates to our second assertion using the Lie bracket (see equation 1.13 on page 17 of [13]). Lastly, "completely integrable" aligns with our first assertion regarding orthogonality conditions (refer to equation 1.16 on page 23 of [13]).

**Theorem E.1** (Frobenius theorem). *Consider $W \subseteq \mathcal{X}(\Omega)$, and assume that the dimension of $W(\theta)$ is constant on $\Omega \subseteq \mathbb{R}^D$. Then the two following assertions are equivalent:*

1. *each $\theta \in \Omega$ admits a neighborhood $\Omega'$ such that there exists $D - \dim(W(\theta))$ independent conserved functions through $W_{|\Omega'}$;*

2. *the following property holds:*

$$[u, v](\theta) \in W(\theta), \quad \text{for each } u, v \in W, \theta \in \Omega \tag{21}$$

**Proposition E.2.** *Under the assumption that* $\dim(W(\theta))$ *is locally constant on* $\Omega$*, Condition* (21) *of Frobenius Theorem holds if, and only if, the linear space* $W' \coloneqq \{\chi \in \mathcal{X}(\Omega), \forall \theta \in \Omega : \chi(\theta) \in W(\theta)\}$ *(which is a priori infinite-dimensional) is a Lie algebra.*

*Proof.* $\Leftarrow$ If $W'$ is a Lie algebra, then as $W \subset W'$ we get: for all $u, v \in W \subset W', [u, v] \in W'$. Given the definition of $W'$ this means that (21) is satisfied.

$\Rightarrow$ Assuming now that (21) holds, we prove that $W'$ is a Lie algebra. For this, given $X, Y \in W'$ we wish to show that $[X, Y](\theta) \in W(\theta)$ for every $\theta \in \Omega$.

Given $\theta \in \Omega$, we first reason as in the first step of the proof of Lemma 3.2 to obtain the existence of a neighborhood $\Omega'$ of $\theta$ and of $m \coloneqq \dim(W(\theta'))$ vector fields $\chi_1, \cdots, \chi_m \in W$ such that $(\chi_1(\theta'), \cdots, \chi_m(\theta'))$ is a basis of $W(\theta')$ for each $\theta' \in \Omega$. By definition of $W'$ we have $X(\theta') \in W(\theta')$ and $Y(\theta') \in W(\theta')$ for every $\theta' \in \Omega'$. Thus, there are smooth functions $a_j, b_j$ such that $X(\cdot) = \sum_1^m a_i(\cdot)\chi_i(\cdot)$ and $Y(\cdot) = \sum_1^m b_i(\cdot)\chi_i(\cdot)$ on $\Omega'$, and we deduce by bilinearity of the Lie brackets that $[X, Y](\theta') = \sum_{i,j}[a_i\chi_i, b_j\chi_j](\theta')$ on $\Omega'$. Since $W(\theta)$ is a linear space, we will conclude that $[X, Y](\theta) \in W(\theta)$ if we can show that $[a_i\chi_i, b_j\chi_j](\theta) \in W(\theta)$. Indeed, we can compute

$$[a_i\chi_i, b_j\chi_j] = a_i b_j[\chi_i, \chi_j] + b_j[(\partial a_i)\chi_j]\chi_j - a_i[(\partial b_j)\chi_i]\chi_j$$

where, due to dimensions, both $(\partial a_i)\chi_j$ and $(\partial b_j)\chi_i$ are smooth scalar-valued functions. By construction of the basis $\{\chi_j\}_j$ we have $\chi_i(\theta), \chi_j(\theta) \in W(\theta)$, and by assumption (21) we have $[\chi_i, \chi_j](\theta) \in W(\theta)$, hence we conclude that $[X, Y](\theta) \in W(\theta)$. Since this holds for any choice of $X, Y \in W'$, this establishes that $W'$ is a Lie algebra. $\square$

**Theorem E.3.** *If* $\dim(\mathrm{Lie}(W_\phi)(\theta))$ *is locally constant then each* $\theta \in \Omega$ *has a neighborhood* $\Omega'$ *such that there are* $D - \dim(\mathrm{Lie}(W_\phi)(\theta))$ *(and no more) independent conserved functions through* $W_{\phi|\Omega'}$*.*

*Proof. 1st step: Existence of $\Omega'$ and of $D - \dim(\mathrm{Lie}(W_\phi)(\theta))$ independent conserved functions.* Let $\theta \in \Omega$. Since $\dim(\mathrm{Lie}(W_\phi)(\theta))$ is locally constant there is a neighborhood $\Omega''$ of $\theta$ on which it is constant. Since $W \coloneqq \mathrm{Lie}(W_\phi)_{|\Omega''} \subseteq \mathcal{X}(\Omega'')$ is a Lie Algebra, by Proposition E.2 and Frobenius theorem (Theorem E.1) there exists a neighborhood $\Omega' \subseteq \Omega''$ of $\theta$ and $D - \dim(W(\theta))$ independent conserved functions through $W_{|\Omega'}$. As $W_\phi \subset \mathrm{Lie}(W_\phi)$, these functions are (locally) conserved through $W_\phi$ too. We only need to show that there are no more conserved functions.

*2nd step: There are no more conserved functions.* By contradiction, assume there exists $\theta_0 \in \Omega$, an open neighborhood $\Omega'$ of $\theta_0$, a dimension $k < \dim(\mathrm{Lie}(W_\phi)(\theta_0))$, and a collection of $D - k$ independent conserved functions through $W_\phi$, gathered as the coordinates of a vector-valued function $h \in \mathcal{C}^1(\Omega', \mathbb{R}^{D-k})$. Consider $W \coloneqq \{X \in \mathcal{X}(\Omega'), \forall \theta \in \Omega', X(\theta) \in \ker \partial h(\theta)\}$. By the definition of independent conserved functions, the rows of the $(D - k) \times D$ Jacobian matrix $\partial h(\theta)$ are linearly independent on $\Omega'$, and the dimension of $W(\theta) = \ker \partial h(\theta)$ is constant and equal to $k$ on $\Omega'$. By construction of $W$ and Proposition 2.5, the $D - k$ coordinate functions of $h$ are independent conserved functions through $W$. Thus, by Frobenius Theorem (Theorem E.1) and Proposition E.2, $W$ is a Lie algebra. By Proposition 2.5 we have $W_\phi(\theta) = \mathrm{range} \partial \phi(\theta)^\top \subset \ker \partial h(\theta)$ on $\Omega'$, hence $W_{\phi|\Omega'} \subset W$, and therefore $\mathrm{Lie}(W_\phi)_{|\Omega'} = \mathrm{Lie}(W_{\phi|\Omega'}) \subset W$. In particular: $\mathrm{Lie}(W_\phi)(\theta_0) \subset W(\theta_0)$, which leads to the claimed contradiction that $\dim(\mathrm{Lie}(W_\phi)(\theta_0)) \leq \dim(W(\theta_0)) = k$. $\square$

# F  Proofs of the Examples of Section 3.3 and additional example

## F.1  Proof of the result given in Example 3.5

**Proposition F.1.** *Consider* $\theta = (U, V) \in \mathbb{R}^{n \times r} \times \mathbb{R}^{m \times r}$*,* $\phi$*, and* $\Omega \subseteq \mathbb{R}^D$*,* $D = (n + m)r$*, as in Example 3.5. The dimension of* $W_\phi(\theta)$ *is constant and equal to* $(n + m - 1)r$ *and* $W_\phi$ *verifies condition* (13) *of Frobenius Theorem (i.e. condition* (21) *of Theorem E.1).*

*Proof.* Denoting $u_i$ (resp. $v_i$) the columns of $U$ (resp. of $V$), for $\theta \in \Omega$ we can write $\phi(\theta) = (\psi(u_i, v_i))_{i=1, \cdots r}$ with $\psi : (u \in \mathbb{R}^n - \{0\}, v \in \mathbb{R}^m - \{0\}) \mapsto uv^\top \in \mathbb{R}^{n \times m}$. As this decouples $\phi$ into $r$ functions each depending on a separate block of coordinates, Jacobian matrices and Hessian

matrices are block-diagonal. Establishing condition (21) of Frobenius theorem is thus equivalent to showing it for each block, which can be done by dealing with the case $r = 1$. Similarly, $W_\phi(\theta)$ is a direct sum of the spaces associated to each block, hence it is enough to treat the case $r = 1$ (by proving that the dimension is $n + m - 1$) to obtain that for any $r \geq 1$ the dimension is $r(n + m - 1)$.

*1st step: We show that $W_\phi$ satisfies condition (21) of Frobenius Theorem.* For $u \in \mathbb{R}^n - \{0\}$, $v \in \mathbb{R}^m - \{0\}$ we write $\theta = (u; v) \in \mathbb{R}^D = \mathbb{R}^{n+m}$ and $\phi_{i,j}(\theta) := u_i v_j$ for $i = 1, \cdots, n$ and $j = 1, \cdots, m$. Now $u_i$ and $v_j$ are *scalars* (and no longer columns of $U$ and $V$). Denoting $e_i \in \mathbb{R}^D = \mathbb{R}^{n+m}$ the vector such that all its coordinates are null except the $i$-th one, we have:

$$\nabla \phi_{i,j}(\theta) = v_j e_i + u_i e_{n+j} \in \mathbb{R}^D,$$
$$\partial^2 \phi_{i,j}(\theta) = E_{j+n,i} + E_{i,j+n} \in \mathbb{R}^{D \times D},$$

with $E_{i,j} \in \mathbb{R}^{D \times D}$ the one-hot matrix with the $(i, j)$-th entry being 1. Let $i, k \in \{1, \cdots, n\}$ and $j, l \in \{1, \cdots, m\}$.

*1st case:* $(i, j) = (k, l)$ Then trivially $\partial^2 \phi_{i,j}(\theta) \nabla \phi_{k,l}(\theta) - \partial^2 \phi_{k,l}(\theta) \nabla \phi_{i,j}(\theta) = 0$.
*2nd case:* $((i \neq k)$ and $(j \neq l))$ Then

$$[\nabla \phi_{i,j}, \nabla \phi_{k,l}](\theta) = (E_{j+n,i} + E_{i,j+n})(v_l e_k + u_k e_{n+l}) - (E_{l+n,k} + E_{k,l+n})(v_j e_i + u_i e_{n+j}) = 0 - 0.$$

*3d case:* $i = k$ and $j \neq l$. Then as $u \neq 0$, there exists $l' \in \{1, \cdots, n\}$ such that $u_{l'} \neq 0$.

$$\partial^2 \phi_{i,j}(\theta) \nabla \phi_{k,l}(\theta) - \partial^2 \phi_{k,l}(\theta) \nabla \phi_{i,j}(\theta) = v_l e_{n+j} - v_j e_{n+l}$$
$$= \frac{v_l}{u_{l'}} \nabla \phi_{l',j}(\theta) - \frac{v_j}{u_{l'}} \nabla \phi_{l',l}(\theta),$$
$$\in \text{span}\{\nabla \phi_{i,j}(\theta)\} = W_\phi(\theta).$$

*4d case:* $((i \neq k)$ and $(j = l))$ We treat this case in the exact same way than the 3d case.

Thus $W_\phi$ verifies condition (13) of Frobenius Theorem.

*2d step: We show that $\dim(W_\phi(\theta)) = (n + m - 1)$.* As $u, v \neq 0$ each of these vectors has at least one nonzero entry. For simplicity of notation, and without loss of generality, we assume that $u_1 \neq 0$ and $v_1 \neq 0$. It is straightforward to check that $(\nabla \phi_{1,1}(\theta), (\nabla \phi_{1,j}(\theta))_{j=2,\cdots,m}, (\nabla \phi_{i,1}(\theta))_{i=2,\cdots,n}$ are $n + m - 1$ linearly independent vectors. To show that $\dim(W_\phi(\theta)) = (n + m - 1)$ is it thus sufficient to show that they span $W_\phi(\theta)$. This is a direct consequence of the fact that, for any $i, j$, we have

$$\nabla \phi_{i,j}(\theta) = v_j e_i + u_i e_{n+j} = \frac{v_j}{v_1}(v_1 e_i + u_i e_{n+1}) + \frac{u_i}{u_1}(u_1 e_{n+j} + v_j e_1) - \frac{v_j u_i}{u_1 v_1}(u_1 e_{n+1} + v_1 e_1),$$
$$= \frac{v_j}{v_1} \nabla \phi_{i,1}(\theta) + \frac{u_i}{u_1} \nabla \phi_{1,j}(\theta) + \frac{v_j u_i}{u_1 v_1} \nabla \phi_{1,1}(\theta). \qquad \square$$

### F.2 An additional example beyond ReLU

In complement to Example 3.5, we give a simple example studying a two-layer network with a positively homogeneous activation function, which include the ReLU but also variants such as the leaky ReLU or linear networks.

*Example* F.2 (Beyond ReLU: Neural network with one hidden neuron with a positively homogeneous activation function of degree one). Let $\sigma$ be a positively one-homogeneous activation function. In (6), this corresponds to setting $g(\theta, x) = \sum_{i=1}^r u_i \sigma(\langle v_i, x \rangle) \in \mathbb{R}$. Assuming $\langle v_i, x \rangle \neq 0$ for all $i$ to avoid the issue of potential non-differentiability at 0 of $\sigma$ (for instance for the ReLU), and in particular assuming $v_i \neq 0$, the function minimized during training can be factored via $\phi(\theta) = (\psi(u_i, v_i))_{i=1}^r$ where

$$\theta := (u \in \mathbb{R}, v \in \mathbb{R}^{d-1} - \{0\}) \overset{\psi}{\mapsto} (u\|v\|, v/\|v\|) \in \mathbb{R} \times \mathcal{S}_{d-1} \subset \mathbb{R}^d. \qquad (22)$$

*Proposition* F.3. *Consider $d \geq 2$ and $\phi(\theta) = (\psi(u_i, v_i))_{i=1}^r$ where $\psi$ is given by (22) on $\Omega := \{\theta = (u \in \mathbb{R}^r, V = (v_1, \ldots, v_r) \in \mathbb{R}^{m \times r}) : v_i \neq 0\}$. We have $\dim(W_\phi(\theta)) = r(d - 1)$ and $W_\phi$ verifies condition (21) of Frobenius Theorem (Theorem E.1), so each $\theta = (u, V) \in \Omega$ admits a neighborhood $\Omega'$ such that there exists $r$ (and no more) conserved function through $W_{\phi|\Omega'}$.*

As in Example 3.5, such candidate functions are given by $h_i : (u_i, v_i) \mapsto u_i^2 - \|v_i\|^2$. A posteriori, these functions are in fact conserved through all $W_\phi$.

*Proof of Proposition F.3.* As in the proof of Proposition F.1 it is enough to prove the result for $r = 1$ hidden neuron. Note that here $D = d$. To simplify notations, we define $\phi_0, ..., \phi_{d-1}$ for $\theta = (u, v)$ as:

$$\phi_0(\theta) = u\|v\|,$$

and for $i = 1, ..., d - 1$:

$$\phi_i(\theta) = v_i/\|v\|.$$

*1st step: explicitation of span$\{\nabla\phi_0, ..., \nabla\phi_{d-1}\}$.* We have

$$\partial\phi(\theta) = \left(\begin{array}{c|c} \|v\| & uv^\top/\|v\| \\ \hline 0_{(d-1)\times 1} & \frac{1}{\|v\|}P_v \end{array}\right),$$

where: $P_v := \mathrm{I}_{d-1} - vv^\top/\|v\|^2$ is the orthogonal projector on $(\mathbb{R}v)^\perp$ (seen here as a subset of $\mathbb{R}^{d-1}$) and its rank is $d - 2$. Thus $\dim(W_\phi(\theta)) = \mathrm{rank}(\partial\phi(\theta)) = d - 1$ and span$\{\nabla\phi_0, ..., \nabla\phi_{d-1}\} = \mathbb{R}\nabla\phi_0 + (\mathbb{R}v)^\perp$.

*2d step: calculation of the Hessians.*

*1st case: The Hessian of $\phi_i$ for $i \geq 1$.* In this case, $\phi_i$ does not depend on the first coordinate $u$ so we proceed as if the ambient space here was $\mathbb{R}^{d-1}$. We have already that for $i \geq 1$:

$$\nabla\phi_i(\theta) = e_i/\|v\| - v_iv/\|v\|^3$$

hence

$$\partial^2\phi_i = 3v_ivv^\top/\|v\|^5 - 1/\|v\|^3\left(v_i\mathrm{I}_{d-1} + V_i + V_i^\top\right),$$

where all columns of matrix $V_i := (0, ..., v, 0, ..., 0)$ are zero except the $i$-th one, which is set to $v$.

*2d case: The Hessian of $\phi_0$.* Since

$$\nabla\phi_0(\theta) = \left(\|v\|, uv^\top/\|v\|\right)^\top.$$

we have

$$\partial^2\phi_0(\theta) = \left(\begin{array}{c|c} 0 & v^\top/\|v\| \\ \hline v/\|v\| & \frac{u}{\|v\|}P_v \end{array}\right).$$

*3rd step: Conclusion.*

*1st case: $i, j \geq 1$ and $i \neq j$.* We have:

$$\partial^2\phi_i(\theta)\nabla\phi_j(\theta) - \partial^2\phi_j(\theta)\nabla\phi_i(\theta),$$
$$= v_j/\|v\|^4 e_i - v_i/\|v\|^4 e_j \in (\mathbb{R}v)^\perp,$$
$$\subset \mathrm{span}\{\nabla\phi_0(\theta), ..., \nabla\phi_{d-1}(\theta)\}.$$

*2d case: $i \geq 1$ and $j = 0$.* We have:

$$\partial^2\phi_i(\theta)\nabla\phi_0(\theta) - \partial^2\phi_0(\theta)\nabla\phi_i(\theta),$$
$$= -2u/\|v\|\nabla\phi_i(\theta),$$
$$\in \mathrm{span}\{\nabla\phi_0(\theta), ..., \nabla\phi_{d-1}(\theta)\}. \qquad\square$$

In both cases, we obtain as claimed that the condition (21) of Frobenius Theorem is satisfied, and we conclude using the latter.

# G   Proof of Proposition 3.7 and additional example

**Proposition G.1.** *Assume that* $\mathrm{rank}(\partial\phi(\theta))$ *is constant on* $\Omega$ *and that* $W_\phi$ *satisfies* (13). *If* $t \mapsto \theta(t)$ *satisfies the ODE* (2) *then there is* $0 < T^\star_{\theta_{\mathrm{init}}} < T_{\theta_{\mathrm{init}}}$ *such that* $z(t) := \phi(\theta(t)) \in \mathbb{R}^d$ *satisfies the ODE*

$$\begin{cases} \dot{z}(t) & = -M(z(t), \theta_{init})\nabla f(z(t)) \quad \text{for all } 0 \leq t < T^\star_{\theta_{\mathrm{init}}}, \\ z(0) & = \phi(\theta_{init}), \end{cases} \tag{23}$$

*where* $M(z(t), \theta_{init}) \in \mathbb{R}^{d\times d}$ *is a symmetric positive semi-definite matrix.*

*Proof.* As $z = \phi(\theta)$ and as $\theta$ satisfies (2), we have:

$$\dot{z} = \partial\phi(\theta)\dot{\theta} = -\partial\phi(\theta)\nabla(f \circ \phi)(\theta) = -\partial\phi(\theta)[\partial\phi(\theta)]^\top \nabla f(z).$$

Thus, we only need to show $M(t) := \partial\phi(\theta(t))[\partial\phi(\theta(t))]^\top$, which is a symmetric, positive semi-definite $d \times d$ matrix, only depends on $z(t)$ and $\theta_{\text{init}}$. Since $\dim W_\phi(\theta) = \texttt{rank}(\partial\phi(\theta))$ is constant on $\Omega$ and $W_\phi$ satisfies (13), by Frobenius Theorem (Theorem E.1), for each $\theta \in \Omega$, there exists a neighborhood $\Omega_1$ of $\theta$ and $D - d'$ independent conserved functions $h_{d'+1}, \cdots, h_D$ through $(W_\phi)_{|\Omega'}$, with $d' := \dim W_\phi(\theta) = \texttt{rank}(\partial\phi(\theta))$. Moreover, by definition of the rank, for the considered $\theta$, there exists a set $I \subset \{1,\ldots,d\}$ of $d'$ indices such that the gradient vectors $\nabla\phi_i(\theta)$, $i \in I$ are linearly independent. By continuity, they stay linearly independent on a neighborhood $\Omega_2$ of $\theta$. Let us denote $P_I$ the restriction to the selected indices and

$$\theta' \in \mathbb{R}^D \longmapsto \Phi_I(\theta') := (P_I\phi(\theta'), h_{d'+1}(\theta'), ..., h_D(\theta')) \in \mathbb{R}^D$$

As the functions $h_i$ are *independent* conserved functions, for each $\theta' \in \Omega' := \Omega_1 \cap \Omega_2$ their gradients $\nabla h_i(\theta')$, $d' + 1 \leq i \leq D$ are both linearly independent and (by Proposition 2.5 and (8)) orthogonal to $W_\phi(\theta') = \text{range}[\partial\phi(\theta')]^\top = \text{span}\{\nabla\phi_i(\theta) : i \in I\}$. Hence, on $\Omega'$, the Jacobian $\partial\Phi_I$ is an invertible $D \times D$ matrix. By the implicit function theorem, the function $\Phi_I$ is thus locally invertible. Applying this analysis to $\theta = \theta(0)$ and using that $h_i$ are conserved functions, we obtain that in an interval $[0, T^\star_{\theta_{\text{init}}})$ we have

$$\Phi_I(\theta(t)) = (P_I z(t), h_{d+1}(\theta_{\text{init}}), ..., h_D(\theta_{\text{init}})) \tag{24}$$

By local inversion of $\Phi_I$ this allows to express $\theta(t)$ (and therefore also $M(t) = \partial\phi(\theta(t))[\partial\phi(\theta(t))]^\top$) as a function of $z(t)$ and of the initialization. $\qquad\square$

In complement to Example 3.8 we provide another example related to Example F.2.

*Example* G.2. Given the reparametrization $\phi : (u \in \mathbb{R}, v \in \mathbb{R}^{d-1} - \{0\}) \mapsto (u\|v\|, v/\|v\|) \in \mathbb{R} \times \mathcal{S}_{d-1} \subset \mathbb{R}^d$ (cf (22)), the variable $z := (r, h) = (u\|v\|, v/\|v\|)$ satisfies (23) with:

$$M(z, \theta_{\text{init}}) = \begin{pmatrix} \sqrt{r^2 + \delta^2} & 0_{1 \times k} \\ 0_{(d-1) \times 1} & \frac{1}{\delta + \sqrt{r^2 + \delta^2}} P_h \end{pmatrix}, \text{ where } P_h := \mathrm{I}_{d-1} - hh^T/\|h\|^2 \text{ and } \delta :=$$

$u^2_{\text{init}} - \|v_{\text{init}}\|^2.$

## H   Proofs of results of Section 4

### H.1   Proof of Proposition 4.2

**Proposition H.1.** *Consider* $\Psi : (U, V) \mapsto U^\top U - V^\top V \in \mathbb{R}^{r \times r}$ *and assume that* $(U; V)$ *has full rank. Then:*

1. *if* $n + m \leq r$, *the function* $\Psi$ *gives* $(n + m)(r - 1/2(n + m - 1))$ *independent conserved functions,*

2. *if* $n + m > r$, *the function* $\Psi$ *gives* $r(r + 1)/2$ *independent conserved functions.*

*Proof.* Let write $U = (U_1; \cdots; U_r)$ and $V = (V_1; \cdots; V_r)$ then: $\Psi_{i,j}(U, V) = \langle U_i, U_j \rangle - \langle V_i, V_j \rangle$ for $i, j = 1, \cdots, r$. Then $f_{i,j} := \nabla\Psi_{i,j}(U, V) = (0; \cdots; 0; \underset{(i)}{U_j}; \cdots; \underset{(j)}{U_i}; 0; \cdots; \underset{(i+r)}{-V_j}; \cdots; \underset{(j+r)}{V_i}; \cdots; 0)^\top \in \mathbb{R}^{(n+m)r \times 1}.$

*1st case:* $n + m \leq r$. As $(U; V)$ has full rank, its rank is $n + m$. In particular, $U$ and $V$ have a full rank too. Without loss of generality we can assume that $(U_1, \cdots, U_{n+m})$ are linearly independent, and $(V_1, \cdots, V_{n+m})$ too. Then for all $i > n + m, U_i \in \mathcal{F}_U := \text{span}(U_1, \cdots, U_{n+m})$ and $V_i \in \mathcal{F}_V := \text{span}(V_1, \cdots, V_{n+m})$. We want to count the number of $f_{i,j}$ that are linearly independent.

1. if $i \leq j \in [\![1, n + m]\!]$, then all the associated $f_{i,j}$ are linearly independent together. There are $(n + m)(n + m + 1)/2$ such functions. Moreover, these functions generate vectors of the form:

$$(A_1; \cdots; A_{n+m}; 0; \cdots; 0; B_1; \cdots; B_{n+m}; 0; \cdots; 0)$$

where $A_i \in \mathcal{F}_U$ and $B_i \in \mathcal{F}_V$.

2. if $i \in [\![1, n+m]\!]$ and $j \in [\![n+m+1, r]\!]$, then all of the associated $f_{i,j}$ are linearly independent and the last ones are linearly independent together. We obtain $(n+m)(r-(n+m))$ more functions. Moreover, these functions generate vectors of the form:

$$(0 \cdots ; 0; A_{n+m+1}; \cdots ; A_r;$$
$$0; \cdots ; 0; B_{n+m+1}; \cdots ; B_r)$$

where $A_i \in \mathcal{F}_U$ and $B_i \in \mathcal{F}_V$.

3. if $i \leq j \in [\![n+m+1, r]\!]$, the associated $f_{i,j}$ are linearly dependent of thus already obtained.

Finally there are exactly $(n+m)(r - 1/2(n+m-1))$ independent conserved functions given by $\Psi$.

*2d case:* $n + m > r$. Then all $(U_i; -V_i)$ for $i = 1, \cdots r$ are linearly independent. Then there are $r(r+1)/2$ independent conserved functions given by $\Psi$. $\qquad\square$

## H.2 Proofs of other results

**Proposition H.2.** *For every* $\Delta \in \mathbb{R}^{n \times m}$ *denote* $S_\Delta := \begin{pmatrix} 0 & \Delta \\ \Delta^\top & 0 \end{pmatrix}$, *one has* $\partial\phi(U, V)^\top : \Delta \in \mathbb{R}^{n \times m} \mapsto S_\Delta \cdot (U; V)$. *Hence* $W_\phi = \mathrm{span}\{A_\Delta, \forall \Delta \in \mathbb{R}^{n \times m}\}$, *where* $A_\Delta : (U; V) \mapsto S_\Delta \cdot (U; V)$ *is a linear endomorphism. Moreover one has* $[A_\Delta, A_{\Delta'}] : (U, V) \mapsto [S_\Delta, S_{\Delta'}] \times (U; V)$.

This proposition enables the computation of the Lie brackets of $W_\phi$ by computing the Lie bracket of matrices. In particular, $\mathrm{Lie}(W_\phi)$ is necessarily of finite dimension.

**Proposition H.3.** *The Lie algebra* $\mathrm{Lie}(W_\phi)$ *is equal to*

$$\left\{ (U; V) \mapsto \begin{pmatrix} \mathrm{I}_n & 0 \\ 0 & -\mathrm{I}_m \end{pmatrix} \times M \times \begin{pmatrix} U \\ V \end{pmatrix} : M \in \mathcal{A}_{n+m} \right\}$$

*where* $\mathcal{A}_{n+m} \subset \mathbb{R}^{(n+m) \times (n+m)}$ *is the space of skew symmetric matrices.*

*Remark* H.4. By the characterization of $\mathrm{Lie}(W_\phi)$ in Proposition H.3 we have that the dimension of $\mathrm{Lie}(W_\phi)$ is equal to $(n+m) \times (n+m-1)/2$.

*Proof.* *1st step:* Let us characterize $W_1 = \mathrm{span}\{W_\phi + [W_\phi, W_\phi]\}$. Let $\Delta, \Delta' \in \mathbb{R}^{n \times m}$, then:

$$[A_\Delta, A_{\Delta'}]((U, V)) = [S_\Delta, S_{\Delta'}] \times (U; V) = \begin{pmatrix} Y, 0 \\ 0, Z \end{pmatrix} \times \begin{pmatrix} U \\ V \end{pmatrix}, \qquad (25)$$

with $Y := \Delta\Delta'^\top - \Delta'\Delta^\top \in \mathcal{A}_n$ and $Z := \Delta^\top\Delta' - \Delta'^\top\Delta \in \mathcal{A}_m$. Then:

$$W_1 = \left\{ (U; V) \mapsto \begin{pmatrix} Y, X \\ X^\top, Z \end{pmatrix} \times \begin{pmatrix} U \\ V \end{pmatrix} : X \in \mathbb{R}^{n \times m}, Y \in \mathcal{A}_n, Z \in \mathcal{A}_m \right\},$$

$$= \left\{ u_M := (U; V) \mapsto \begin{pmatrix} \mathrm{I}_n, 0 \\ 0, -\mathrm{I}_m \end{pmatrix} \times M \times \begin{pmatrix} U \\ V \end{pmatrix} : M \in \mathcal{A}_{n+m} \right\}.$$

*2d step:* Let us show that $W_2 = W_1$. Let $M, M' \in \mathcal{A}_{n+m}$. Then:

$$[u_M, u_{M'}] = \begin{pmatrix} \mathrm{I}_n, 0 \\ 0, -\mathrm{I}_m \end{pmatrix} \left( M \begin{pmatrix} \mathrm{I}_n, 0 \\ 0, -\mathrm{I}_m \end{pmatrix} M' - M' \begin{pmatrix} \mathrm{I}_n, 0 \\ 0, -\mathrm{I}_m \end{pmatrix} M \right) = \begin{pmatrix} \mathrm{I}_n, 0 \\ 0, -\mathrm{I}_m \end{pmatrix} \tilde{M},$$

with $\tilde{M} := M \begin{pmatrix} \mathrm{I}_n, 0 \\ 0, -\mathrm{I}_m \end{pmatrix} M' - M' \begin{pmatrix} \mathrm{I}_n, 0 \\ 0, -\mathrm{I}_m \end{pmatrix} M \in \mathcal{A}_{n+m}$.

*Finally:* $\mathrm{Lie}(W_\phi) = W_1 = \left\{ (U; V) \mapsto \begin{pmatrix} \mathrm{I}_n, 0 \\ 0, -\mathrm{I}_m \end{pmatrix} \times M \times \begin{pmatrix} U \\ V \end{pmatrix} : M \in \mathcal{A}_{n+m} \right\}.$ $\qquad\square$

Eventually, what we need to compute is the dimension of the trace $\mathrm{Lie}(W_\phi)(U, V)$ for any $(U, V)$.

**Proposition H.5.** *Let us assume that* $(U; V) \in \mathbb{R}^{(n+m) \times r}$ *has full rank. Then:*

*1. if $n + m \leq r$, then* $\dim(\mathrm{Lie}(W_\phi)(U;V)) = (n+m)(n+m-1)/2$;

*2. if $n + m > r$, then* $\dim(\mathrm{Lie}(W_\phi)(U;V)) = (n+m)r - r(r+1)/2$.

*Proof.* Let us consider the linear application:

$$\Gamma : M \in \mathcal{A}_{n+m} \mapsto \begin{pmatrix} \mathrm{I}_n, 0 \\ 0, -\mathrm{I}_m \end{pmatrix} \times M \times \begin{pmatrix} U \\ V \end{pmatrix},$$

where $\mathcal{A}_{n+m} \subset \mathbb{R}^{(n+m)^2}$ is the space of skew symmetric matrices. As $\mathrm{range}\Gamma(\mathcal{A}_{n+m}) = \mathrm{Lie}(W_\phi)(U;V)$, we only want to calculate $\mathrm{rank}\Gamma(\mathcal{A}_{n+m})$. But by rank–nullity theorem, we have:

$$\dim \ker \Gamma + \mathrm{rank} \ \Gamma = (n+m)(n+m-1)/2.$$

*1st case: $n + m \leq r$.* Then as $(U;V)$ has full rank $n + m$, $\Gamma$ is injective and then $\mathrm{rank}\Gamma(\mathcal{A}_{n+m}) = (n+m)(n+m-1)/2$.

*2d case: $n + m > r$.* We write $(U;V) = (C_1; \cdots ; C_r)$ with $(C_1, \cdots, C_r)$ that are linearly independent as $(U;V)$ has full rank $r$. Let $M \in \mathcal{A}_{n+m}$ such that $\Gamma(M) = 0$. Then $M \cdot (U;V) = 0$. Then we write $M^\top = (M_1; \cdots ; M_{n+m})$. Then as $M \times (U;V) = 0$, we have that $\langle M_i, C_j \rangle = 0$ for all $i = 1, \cdots, n + m$ and for all $j = 1, \cdots, r$. We note $C := \mathrm{span}_{i=1,\cdots,r} C_i$ that is of dimension $r$ as $(U;V)$ has full rank $r$.

$M_1$ must be in $C^\perp$ and its first coordinate must be zero as $M$ must be a skew matrix. Then $M_1$ lies in a space of dimension $n + m - r - 1$. Then $M_2$ must be in $C^\perp$ too, and its first coordinate is determined by $M_1$ and its second is null as $M$ is a skew matrix. Then $M_2$ lies in a space of dimension $n + m - r - 2$. By recursion, after building $M_1, \cdots, M_i$, $M_{i+1}$ must be in $C^\perp$ too, and its $i$ first coordinates are determined by $M_1, \cdots, M_i$ and its $i + 1$-th one is null as $M$ is a skew matrix. Then $M_{i+1}$ lies in a space of dimension $\max(0, n + m - r - (i+1))$. Finally the dimension of $\ker\Gamma$ is equal to:

$$\sum_{i=1}^{n+m-r} (n + m - r - i) = (n + m - r - 1)(n + m - r)/2.$$

Then: $\mathrm{rank}\Gamma(\mathcal{A}_{n+m}) = (n+m)r - r(r+1)/2$. $\qquad\square$

Thanks to this explicit characterization of the trace of the generated Lie algebra, combined with Proposition 4.2, we conclude that Proposition 4.1 has indeed exhausted the list of independent conservation laws.

**Corollary H.6.** *If $(U;V)$ has full rank, then all conserved functions are given by $\Psi : (U, V) \mapsto U^\top U - V^\top V$. In particular, there exist no more conserved functions.*

*Proof.* As $(U;V)$ has full rank, this remains locally the case. By Proposition 4.3 the dimension of $\mathrm{Lie}(W_\phi)(U;V)$ is locally constant, denoted $m(U,V)$. By Theorem 3.3, the exact number of independent conserved functions is equal to $(n+m)r - m(U,V)$ and that number corresponds to the one given in Proposition 4.2. $\qquad\square$

# I  About Example 3.6

**Proposition I.1.** *Let us assume that $(U;V) \in \mathbb{R}^{(n+m)\times r}$ has full rank. If $\max(n, m) > 1$ and $r > 1$, then $W_\phi$ does not satisfy the condition (13).*

*Proof.* Let us consider the linear application:

$$\Gamma' : \Delta \in \mathbb{R}^{n \times m} \mapsto \begin{pmatrix} 0, \Delta \\ \Delta^\top, 0 \end{pmatrix} \times \begin{pmatrix} U \\ V \end{pmatrix}.$$

By Proposition H.2, $\mathrm{range}\Gamma'(\mathbb{R}^{n \times m}) = W_\phi(U;V)$. Thus, as by definition $W_\phi(U;V) \subseteq \mathrm{Lie}(W_\phi(U;V))$, $W_\phi$ does not satisfy the condition (13) if and only if $\dim(W_\phi(U;V)) < \dim(\mathrm{Lie}W_\phi(U;V))$.

*1st case:* $n + m \leq r$. Then as $(U; V)$ has full rank $n + m$, $\Gamma'$ is injective and then $\text{rank} \Gamma'(\mathbb{R}^{n \times m}) = n \times m$.

Thus by Proposition H.5, we only need to verify that: $n \times m < (n + m)(n + m - 1)/2 =: \text{Lie} W_\phi(U; V)$. It is the case as $\max(n, m) > 1$.

*2d case:* $n + m > r$. We write $(U; V) = (C_1; \cdots; C_r)$ with $(C_1, \cdots, C_r)$ that are linearly independent as $(U; V)$ has full rank $r$. Let $\Delta \in \mathbb{R}^{n \times m}$ such that $\Gamma'(\Delta) = 0$. Let us define the symmetric matrix $M$ by:

$$M := \begin{pmatrix} 0, \Delta \\ \Delta^\top, 0 \end{pmatrix}. \tag{26}$$

Then $M \cdot (U; V) = 0$. Then we write $M^\top = (M_1; \cdots; M_{n+m})$. Then as $M \times (U; V) = 0$, we have that $\langle M_i, C_j \rangle = 0$ for all $i = 1, \cdots, n + m$ and for all $j = 1, \cdots, r$. We note $C := \underset{i=1,\cdots,r}{\text{span}} \, C_i$

that is of dimension $r$ as $(U; V)$ has full rank $r$.

For all $i = 1, \cdots, n$, $M_i$ must be in $C^\perp$ and its $n$ first coordinate must be zero by definition (26). Then $M_i$ lies in a space of dimension $\max(0, n + m - r - n)$. For all $j > n$, $M_j$ are entirely determined by $\{M_i\}_{i \leq n}$ by definition (26). Finally the dimension of $\ker \Gamma'$ is equal to: $n \times \max(0, m - r)$. Then: $\dim(W_\phi(U; V)) = \text{rank} \Gamma'(\mathbb{R}^{n \times m}) = nm - n \times \max(0, m - r)$.

Thus by Proposition H.5, we only need to verify that: $nm - n \max(0, m - r) < (n + m)r - r(r + 1)/2 =: \text{Lie} W_\phi(U; V)$.

*Let us assume* $m < r$. Then by looking at $f(r) := (n + m)r - r(r + 1)/2 - nm = \dim(\text{Lie} W_\phi(U; V)) - \dim(W_\phi(U; V))$ for $r \in \{m + 1, \cdots, n + m - 1\} =: I_{n,m}$, we have: $f'(r) = (n + m) - 1/2 - r > 0$ (as $n + m > r$ is an integer), so $f$ is increasing, so on $I_{n,m}$, we have (as $r > m$): $f(r) > f(m) = (n + m)m - m(m + 1)/2 - nm = m^2 - m(m + 1)/2 \geq 0$ as $m \geq 1$.

*Let us assume* $m \geq r$. Then

$$\begin{aligned}
\dim(\text{Lie} W_\phi(U; V)) - \dim(W_\phi(U; V)) &= (n + m)r - r(r + 1)/2 - (nm - n(m - r)), \\
&= mr - r(r + 1)/2, \\
&\geq r^2 - r(r + 1)/2 \quad \text{as } m \geq r, \\
&> 0 \quad \text{as } r > 1.
\end{aligned}$$

Thus $\dim(\text{Lie} W_\phi(U; V)) - \dim(W_\phi(U; V)) > 0$. $\qquad \square$

## J  Details about experiments

We used the software SageMath [29] that relies on a Python interface. Computations were run in parallel using 64 cores on an academic HPC platform.

First we compared the dimension of the generated Lie algebra $\text{Lie}(W_\phi)(\theta)$ (computed using the algorithm presented in Section 3.3) with $D - N$, where $N$ is the number of independent conserved functions known by the literature (predicted by Proposition 4.1 for ReLU and linear neural networks). We tested both linear and ReLU architectures (with and without biases) of various depths and widths, and observed that the two numbers matched in all our examples.

For this, we draw 50 random ReLU (resp. linear) neural network architectures, with depth drawn uniformly at random between 2 to 5 and i.i.d. layer widths drawn uniformly at random between 2 to 10 (resp. between 2 to 6). For ReLU architectures, the probability to include biases was $1/2$.

Then we checked that all conservation laws can be explicitly computed using the algorithm presented in Section 2.5 and looking for polynomial solutions of degree 2 (as conservation laws already known by the literature are polynomials of degree 2). As expected we found back all known conservation laws by choosing 10 random ReLU (resp. linear) neural network architectures with depth drawn uniformly at random between 2 to 4 and i.i.d. layer widths drawn uniformly at random between 2 to 5.