# OpenReview forum: "Abide by the law and follow the flow: conservation laws for gradient flows"
_NeurIPS.cc/2023/Conference — NeurIPS 2023 oral_

### Official Review · Reviewer_pz1Q · 2023-07-03

**Soundness:** 3 good
**Presentation:** 3 good
**Contribution:** 2 fair
**Rating:** 7
**Confidence:** 3

**Summary:**

The authors study conservation laws in the gradient flow dynamics of neural networks.
They introduce a notion of local factorisation of the loss, stating that the loss in the neighbourhood of a given wieght vector can be decomposed in a composition of functions, a data-independent term $\phi$ followed by a data dependent one $f$.

Under this decomposition, the authors set out to characterise conserved quantities for fixed $\phi$ and for all $f$ giving rise to a proper ERM loss.
They claim that under some assumption, and for ReLU networks, this is equivalent of characterising conserved quantities for fixed $\phi$ and for all smooth $f$.
More strongly, they claim that this is also equivalent to characterising conserved quantities for fixed $\phi$ and for a special finite dimensional subspace of function, living in the linear span of the $d$ rows of the Jacobian of $\phi$, where $d$ is the dimension of the codomain of $\phi$.

The authors then notice that the conservation laws of linear and ReLU networks will be polynomial in the weights, and consider the question of characterising maximal sets of conservation laws.
They link such number to the dimension of the Lie algebra generated locally by the $d$-dimensional vector field associated to the Jacobian of $\phi$, and work-out explicitly some examples.

Finally, the authors consider the question whether the known conservation laws for two-layers linear and ReLU networks form a maximal set, and find an affirmative answer.
They conjecture, and briefly discuss this point in Appendix 9, that the same results hold for larger depths.

**Strengths:**

- The article is well written, and guides the reader nicely through quite technical results without requiring too much previous knowledge.
- The results seem novel to the extent of my knowledge of the literature, which is not comprehensive on this subject. They seem to fit nicely into a pre-existing line of works (see ref [13] and [26] for example).
- The main text provides reasonable justification for most of the formal results, which are proven in the Appendix. While I did not check the correctness of the proofs, the results seem reasonable.

**Weaknesses:**


- It is not clear to what extent the factorisation property the authors introduce is necessary to their analysis. Said differently, it is not clear wehther the analysis could be performed similarly on a concrete example (say fixed dataset) without any mention of a factorisation. The authors could add additional high-level explanations/justifications of this concept.

**Questions:**


## Major questions

- It is not clear to me whether the requirement in Eq. 2 is vacuous, meaning that any architecture automatially satisfies it. Can't I just take $\phi$ as the identity map, and $f = \mathcal{E}$? If this trivial factorisation is possible, are the results still non-trivial? Maybe I am losing some very simple nuance. In any case, adding a counter-example, or clarifying better the requirements on $\phi$ and $f$ could avoid doubts.

- Example 2.2 provides only a local factorisation for ReLU networks. Of course locality is ok as we are considering gradient flow dynamics, which is local. But I wonder whether something special can happen at the boundaries of the set $\Omega$ defined in Example 2.2, i.e. if there is some gluing condition/gluing phenomenon that may affect the results presented by the authors.

- Is there a commonly used loss for which Eq. 7 is not satisfied?

- The authors stress that their results allow for explicit construction of maximal sets of conserved quantities, yet in the manuscript they provide only an a posteriori verification that known conserved quantities in previously studied architectures indeed form maximal sets. Is there an architecture where new conservation laws can be found through the presented techniques?

## Suggestions for manuscript improvement

line 50: missing closing bracket

line 106: specifying that $D$ is the number of weigths, and $d$ is the dimension of the "internal representation" of the decomposition $\mathcal{E} = f \circ \phi$ would be helpful for the reader.

line 119: in the inlined equation, one has a tensor $\phi$ with three free indices equal to an expression without the same three free indices. I suggest to clarify this writing somehow.

**Limitations:**

The authors briefly discuss limitations in the main text.
I would add that the analysis seems to be limited to linear and ReLU architectures 2 layer architectures. It is not clear whether a local factorisation of the form Eq. 2 can be found for other architectures. The authors could maybe add a discussion on this point.

---

> ### Author Rebuttal · Authors · 2023-08-08
>
> We would like to thank the reviewer for the positive comments and constructive suggestions.
>
> **Weaknesses and Questions addressed**
>
> > **L1e** It is not clear to what extent the factorisation property the authors introduce is necessary to their analysis. Said differently, it is not clear whether the analysis could be performed similarly on a concrete example (say fixed dataset) without any mention of a factorisation. The authors could add additional high-level explanations/justifications of this concept.
>
> Thank you for the insightful observation. When you fix the dataset and the loss, you're essentially dealing with a singular vector field, specifically: $\theta \mapsto \nabla \mathcal{E} (\theta)$. In this particular scenario, our framework is directly applicable. Given that we're examining a singular vector field, its associated Lie algebra has a dimension of one. This results in $D-1$ conserved quantities. However, it's worth noting that these quantities intricately depend on both the chosen dataset and the specified loss. Consequently, the utility of such an analysis, which is inherently data-dependent, may be somewhat limited. We'll be sure to include this point as a clarifying remark in our work.
>
> > **Q1e** It is not clear to me whether the requirement in Eq. 2 is vacuous, meaning that any architecture automatically satisfies it. Can't I just take $\phi$ as the identity map, and $f = \mathcal{E}$? If this trivial factorisation is possible, are the results still non-trivial? Maybe I am losing some very simple nuance. In any case, adding a counter-example, or clarifying better the requirements on $\phi$ and $f$ could avoid doubts.
>
> Thank you for raising this point! If you adopt this simplistic factorization, then $ \partial \phi (\theta) = I_D$, leading to $V_\phi(\theta) = \mathbb{R}^D$. This means that for this particular $\phi$, there isn't any conservation law. In essence, such a parametrization doesn't have the requisite "tightness" to yield an equivalent of Theorem 2.8. This is a very good remark and we will add it to clarify!
>
> > **Q2e** Example 2.2 provides only a local factorisation for ReLU networks. Of course locality is ok as we are considering gradient flow dynamics, which is local. But I wonder whether something special can happen at the boundaries of the set Ω defined in Example 2.2, i.e. if there is some gluing condition/gluing phenomenon that may affect the results presented by the authors.
>
> In the case of linear / ReLU networks, our analysis shows that there are no conservation laws beyond existing conservation laws, which are known to be global. This addresses the gluing issue for such cases. For more general settings compatible with our analysis, which is indeed only local, gluing conditions are an interesting but possibly difficult challenge, we will comment a bit on this in the revised version.
>
> > **Q3e** Is there a commonly used loss for which Eq. 7 is not satisfied?
>
> As mentioned in Remark A.7, Eq. 7 is not satisfied for cross-entropy loss. However, it is possible to envision weaker assumptions on the span involved in Eq. 7 to extend the theory to such a loss. This is an interesting challenge left for further work.
>
> > **Q4e** The authors stress that their results allow for explicit construction of maximal sets of conserved quantities, yet in the manuscript they provide only an a posteriori verification that known conserved quantities in previously studied architectures indeed form maximal sets. Is there an architecture where new conservation laws can be found through the presented techniques?
>
> In the manuscript, Section 2.4 provides an algorithm (Algo<1>) that constructs directly all polynomial conservation laws. By comparing the number of independent polynomial conservation laws with $D - m$, with $m$ the dimension of the trace of the generated Lie algebra (whose algorithm (Algo<2>) is described in section 3.3), we found that the numbers match and that all polynomial conservation laws found by Algo<1> correspond to the ones already known by the literature. By Algo<2>, we know that there are no other conserved quantities.
>
> The only “a posteriori” reasoning in our analysis is to say that all conserved quantities are in fact global as discussed in our answer to **Q2e**.
>
> Regarding the *discovery* of new conservation laws: for architectures involving piecewise polynomial activations, we expect that a polynomial $\phi$ yielding the factorization $f \circ \phi$ can again be found, allowing to conduct the same analysis but different (polynomial) conservations laws. The main challenge, left to future work, would be to establish an equivalent of Th 2.8.
>
> > **L1e** The authors briefly discuss limitations in the main text. I would add that the analysis seems to be limited to linear and ReLU architectures 2 layer architectures. It is not clear whether a local factorisation of the form Eq. 2 can be found for other architectures. The authors could maybe add a discussion on this point.
>
> For deeper architectures, we still have a local factorization $\phi$ of the form Eq 2, as mentioned in our paper (ll 122-123, appendix A.2, section 4.2). It will be a good challenge to generalize theorem 2.8 to deeper cases, which we will do for further work.

---

> > ### Comment · Reviewer_pz1Q · 2023-08-10
> >
> > I thank the authors for addressing in detail the points I raised in the review.
> > In particular, the authors adequately addressed my main doubts regarding the role of the factorisation introduced in the paper.
> > After assessing their comments, I recognise that the factorisation is indeed non-trivial and bears non-trivial consequences.
> > I decided to update my overall grading to 7 to reflect this.

---

### Official Review · Reviewer_7TLM · 2023-07-04

**Soundness:** 3 good
**Presentation:** 3 good
**Contribution:** 4 excellent
**Rating:** 7
**Confidence:** 4

**Summary:**

This paper studies the conservation law of gradient flow dynamics for training neural networks. The authors propose a method to determine the number of conservation laws in given gradient flow dynamics using Lie algebra generated by the Jacobian vector fields. It is shown, either theoretically or empirically, that the known conservation laws in training linear networks and ReLU networks are also maximal.

**Strengths:**

1. The paper is well-written and clear. The conversation laws in gradient flow dynamics for training neural networks have facilitated the analysis of convergence and implicit bias, thus deserving formal analysis on finding these conversation laws given any network architecture.
2. This paper has an in-depth discussion of the conservation law under the gradient flow on a class of loss functions.
3. Analysis via Lie algebra that determines the maximum number of conservation laws.
4. Showing existing conservation laws studied for linear and ReLU networks are maximal



**Weaknesses:**

none

**Questions:**

Do the results stated for ReLU networks hold for any network with homogeneous activation function?

**Limitations:**

authors discussed the limitation.

---

> ### Author Rebuttal · Authors · 2023-08-08
>
> We would like to thank the reviewer for the positive comments and constructive suggestions.
>
> **Question addressed**
>
> > **Q1d** Do the results stated for ReLU networks hold for any network with homogeneous activation function?
>
> Yes, our results also apply to networks using any positively $p$-homogeneous activation function. Specifically, for a 2-layer NN given by $g_{\theta}(x) = U \sigma(V^\top x)$, the conserved quantities are $\|u_i\|^2 - \|v_i\|^2/p$, where $u_i$ and $v_i$ are the columns of $U$ and $V$, respectively. We plan to include this example in the supplementary material of the final version.

---

### Official Review · Reviewer_Tpzp · 2023-07-05

**Soundness:** 4 excellent
**Presentation:** 3 good
**Contribution:** 3 good
**Rating:** 7
**Confidence:** 3

**Summary:**

The paper discusses the geometric properties of gradient descent dynamics in ML models. The authors aim to understand the properties of the optimization initialization that are preserved during the dynamics, which is often referred to as being an "implicit bias" of the training algorithm. They also focus on the maximal sets of independent quantities conserved during gradient flows. They have an interesting approach to find the exact number of these conserved quantities by performing algebraic manipulations on the Lie algebra generated by the Jacobian of the model.

The paper's contributions include formalizing the notion of a conservation law in the setting of training neural networks, proposing an algorithm to identify simply expressed (e.g., polynomial) conservation laws on ReLU NNs, and illustrating how these findings can rewrite an over-parameterized flow as an "intrinsic" low-dimensional flow. I find it very intriguing that the commonly reported conservation laws in the literature happen to be maximal (at least empirically).

**Strengths:**

The manuscript has many strengths and overall I think this is a welcomed contribution to the literature. The manuscript covers various aspects of gradient dynamics, conserved functions, conservation laws, Lie algebra, and their applications in neural networks with illustrative examples. There is also potentially interesting practical consequences of this work. It seems to be a relatively practical approach for determining the number of conservation laws using Lie Group computations, at least on NNs with piecewise linear activation functions. It is interesting that their algorithms confirm (at least empirically) that the conservation laws for ReLU NNs match the laws already known.

**Weaknesses:**

A few weaknesses are:

- Mostly restricted to deep shallow NNs, continuous-time gradient descent, and simple NN architectures. This limits the applications of the theory to practical situations. The continuous-time restriction on the gradient descent training algorithm is perhaps the more

- In most situations, the generated Lie algebra is going to be infinite-dimensional.  In fact, the two examples in the manuscript are contrived so that the Lie algebra ends up being finite-dimensional. The discussion on the case when the Lie algebra is infinite-dimensional, is only briefly discussed. I would suggest that the author discussed this more. In particular, the stopping criteria are based on the trace of Lie group algebra.

- The above discussion is particularly important for hoping to apply these techniques on NNs with activation functions that are not piecewise linear.

**Questions:**

- Can you add more discussion on the situation where the Lie algebra is infinite-dimensional?

- Can you add more discussion on what happens when the activation functions of the NN are not piecewise linear?

**Limitations:**

There are no potential negative societal impacts of this work.

---

> ### Author Rebuttal · Authors · 2023-08-08
>
> We would like to thank the reviewer for the positive comments and constructive suggestions.
>
> **Weaknesses and Questions addressed**
>
> > **W1c** Mostly restricted to deep shallow NNs, continuous-time gradient descent, and simple NN architectures. This limits the applications of the theory to practical situations. The continuous-time restriction on the gradient descent training algorithm is perhaps the more
>
> Experiments that display the approximate conservation of these laws during the process of gradient descent (as opposed to gradient flow) can be found in Figure 1 of [7] for a 2-layer linear NN, in Figure 1c of [1] for a 3-layer linear NN, and in Figure 2 of [7] for a 3-layer ReLU NN. We plan to include a more detailed commentary on this observation.
>
> In addition to the standard multilayer linear/ReLU architectures discussed in this paper, preliminary studies, beyond the confines of this work, suggest that more diverse ReLU architectures, which encompass aspects like residual connections and convolutional layers, adhere to such a factorization with polynomial $\phi$. With a suitable adaptation of Theorem 2.8, our entire framework should be applicable in these scenarios.
>
> > **W2c** In most situations, the generated Lie algebra is going to be infinite-dimensional. In fact, the two examples in the manuscript are contrived so that the Lie algebra ends up being finite-dimensional. The discussion on the case when the Lie algebra is infinite-dimensional, is only briefly discussed. I would suggest that the author discussed this more. In particular, the stopping criteria are based on the trace of Lie group algebra.
>
> Generally speaking, the Lie algebra generated can indeed be of infinite dimension. This is particularly the case for deeper linear neural networks where $q > 2$, a point we intend to address in the final version. However, the *trace* of the generated Lie algebra is invariably finite-dimensional—it is bounded by $D$, the total number of parameters—and this trace is our focal point when considering the stopping criterion. As an analogy, the set of all smooth real-valued functions constitutes an infinite-dimensional Lie algebra, yet its trace at any given point corresponds to the finite-dimensional space $\mathbb{R}$, thus having a dimensionality of one. Given that the trace has a finite dimension, we can deduce a basis for it. Consequently, the stopping criterion will be met in a maximum of $D$ steps. We will emphasize this distinction in our work.
>
> > **W3c**, **Q1c**, **Q2c** The above discussion is particularly important for hoping to apply these techniques on NNs with activation functions that are not piecewise linear. Can you add more discussion on the situation where the Lie algebra is infinite-dimensional?
> Can you add more discussion on what happens when the activation functions of the NN are not piecewise linear?
>
> Our theory readily accommodates an infinite-dimensional Lie algebra, given that its trace remains finite-dimensional. As an illustration, for deeper linear networks (where $q > 2$), the Lie algebra does become infinite-dimensional. Yet, our theory remains applicable, especially since theorem 2.8 is valid for linear networks irrespective of their depth. We will elucidate this point in the final version. When dealing with more intricate activation functions, our results are directly applicable for any positively $p$-homogeneous activation function. Specifically, for a 2-layer NN represented as $g_{\theta} (x) = U \sigma(V^\top x)$, the conserved quantities are given by $\|u_i\|^2 - \|v_i\|^2/p$, where $u_i$ and $v_i$ denote the columns of $U$ and $V$ respectively. We plan to incorporate this example in the supplementary material of the final version. An intriguing avenue for exploration would be to extend these findings to encompass piecewise linear and piecewise-polynomial activations.

---

### Official Review · Reviewer_wzMU · 2023-07-06

**Soundness:** 4 excellent
**Presentation:** 3 good
**Contribution:** 4 excellent
**Rating:** 7
**Confidence:** 4

**Summary:**

This paper studies the conservation laws, which are quantities that remain constant, in over-parametrized gradient flows. The authors provide a formal definition for independent conserved functions, which are required to have linearly independent gradients. By applying Frobenius theorem, the authors show that the number of independent conservation laws is linked to the dimension of the trace of the Lie algebra generated by the vector fields spanned by the Jacobian. When this vector field is a Lie algebra, Frobenius theorem can be applied directly to obtain the number of independent conservation laws. When the vector field is not a Lie algebra, the generated Lie algebra need to be computed before applying Frobenius theorem. The authors explicitly compute the Lie algebra and its trace for two layer linear networks and certain ReLU networks, obtained the number of independent conservation laws, and prove that the conservation laws discovered in previous literature are complete. They implement their algorithm in SageMath that constructs a basis of polynomial conservation laws for the above examples, and successfully verify the number of independent conservation laws.


**Strengths:**

- This paper is the first to formally define and study the number of independent conservation laws in gradient flow. Previous works mostly focus on finding conserved quantities in different architectures or using them in convergence proofs. This paper provides a new perspective and contributes to a unified framework of conservation laws in gradient flows.
- Using Frobenius theorem to characterize the number of independent conservation laws is novel. By linking to the dimension of the trace of the Lie algebra generated by the Jacobian, the authors present the first known method to determine the number of conservation laws. This method yields the interesting result that known conservation laws are complete in 2-layer linear networks.
- The idea that conserved functions define invariant hyper-surfaces which trap the gradient flow is interesting and useful. Studying the dimension of these surfaces directly leads to the proposed definition of independent conserved functions.
- The authors provide a condition under which gradient flows can be recast as low-dimensional Riemannian flows (proposition 3.8), which has potential applications on choosing initializations for better convergence.


**Weaknesses:**

- The paper’s contribution is overall limited in the aspect of applications. Explicit conservations laws are only given for two-layer linear networks and certain two-layer ReLU networks. The analysis is applied to continuous gradient flow only, and there is no discussion or experiment that verify how well the conservation laws hold in gradient descent. Many neural networks today have more complicated architectures, such as residual connections and various activations other than ReLU, and are often trained with different optimization algorithms, such as Adam. Therefore, while this paper is a promising start to understand implicit bias, more work is needed to obtain insights useful for common machine learning tasks.
- The abstract vaguely mentions “understanding desirable properties of optimization initialization in large machine learning models”, but the paper provides little supporting arguments. It is not clear what the desirable properties are, and whether it is possible to extend the conservation laws to more realistic settings in large models.
- The requirement to factor the cost in equation 2 seems strict - $f$ cannot depend on $\theta$ and $\phi$ cannot depend on the data and the loss $l$. Factorization for two-layer ReLU network (Example 2.2) is a good example that extends beyond linear networks. However, it is not clear whether this is possible with other activation functions, where the pre-activation is not piecewise linear.
- There are a few cases where definitions and theorems are mentioned well before the formal statement, for example, in line 134-135, 156-157, 168, 270, etc. Perhaps the organization could be improved to reduce the complexity of the logic flow.


**Questions:**

- The factorization for two-layer ReLU network requires that $\epsilon_{j,x_i} = \mathbb{1} (v_j^T x_i) > 0$ is constant. How likely does this condition hold throughout the gradient flow?
- Would it be possible to include a brief summary of what the Frobenius theorem is about? This theorem appears to be an important foundation, but the form used in this paper (theorem A.12) appears different from the theorem in the given reference [10] (“Theorem 1.4.1. A nonsingular distribution is completely integrable if and only if it is involutive.”)
- Conservation laws is also an important concept in physics. Is the algorithm that constructs conservation laws related to methods in physics, such as the the Noether’s theorem or conserved quantities from the Killing vector field? Has there been similar analysis on the number of conservation laws for physical systems?

**Limitations:**

The authors included limitations by clearly stating the assumptions. There are no potential negative societal impacts of the work.

---

> ### Author Rebuttal · Authors · 2023-08-08
>
> We would like to thank the reviewer for the positive comments and constructive suggestions.
> > **W1b** The paper’s contribution is overall limited in the aspect of applications [...]
>
> For ReLU/linear networks *of any depth*, explicit conservation laws of $\phi$ are known (Prop 4.1) and our algorithms allow us to verify on a number of *deep (q>2)* ReLU/linear networks that there are no other ones (section 4.2). The only ingredient of our framework that is not yet extended to deeper (q>2) *ReLU* architectures is Theorem 2.8, which ensures that the conservation laws of $\phi$ (computable with our algorithms) are indeed exactly the conservation laws shared by all $\mathcal{E}$, for any dataset and loss. For linear networks, Theorem 2.8 is valid irrespective of their depth.
>
> Experiments showing approximate conservation of these laws during gradient descent (instead of gradient flow) are given in figure 1 of [7] for 2-layer linear NN, figure 1c of [1] for 3-layer linear NN, and figure 2 of [7] for a 3-layer ReLU NN. We will add a more explicit comment on this fact. Regarding other optimization algorithms, while our framework leaves completely open the question of a similar analysis for stochastic algorithms, it seems feasible to adapt it to deterministic algorithms with momentum using their associated ODE. This is however out of scope and left for further work.
>
> Beyond standard multilayer linear/ReLU architectures covered in this work, it seems feasible but technical (and beyond the scope of this paper) to show that more general ReLU architectures covering e.g. residual connections and convolutive layers, satisfy such a factorization with polynomial $\phi$. The main challenge for a follow-up is then to extend Theorem 2.8, which would make the whole framework applicable in such contexts.
>
> For more general activation functions, our results directly apply when using any positively $p$-homogeneous activation function. For a 2-layer NN defined as $g_{\theta} (x) = U \sigma(V^\top x)$, the conserved quantities are given by $\|u_i\|^2 - \|v_i\|^2/p$, where $u_i$ and $v_i$ are the columns of $U$ and $V$ respectively. We plan to incorporate this example in the supplementary material of the final version. A compelling direction for future exploration would be to extend these results to piecewise linear and piecewise-polynomial activations.
>
> > **W2b** The abstract vaguely mentions [...]
>
> In general terms, "desirable properties" of initialization refer to characteristics that ensure convergence, and potentially faster convergence, to an optimal solution. For instance, the utilization of conservation laws in [6] and [*] demonstrates convergence, while [3] illustrates that initializing with certain values of the conserved function can lead to accelerated convergence. We will clarify this further in the final version of the paper. Also, refer to our response to **W1b**.
>
> [*] "On the Convergence of Gradient Descent Training for Two-layer ReLU-networks in the Mean Field Regime" by S. Wojtowytsch, 2020, preprint.
>
> > **W3b** The requirement to factor the cost in eq. 2 seems strict [...]
>
> Eq. 2 is in fact not very demanding: we can always write $\mathcal{E} = f \circ \phi$ with $f= \mathcal{E}$ and $\phi = id$. However the number of conservation laws of $\phi= id$ is zero, and this trivial factorization fails to capture the existence and number of conservation laws as studied in this paper. This suggests that, among all existing factorizations $\mathcal{E} = f \circ \phi$, there may be a notion of an optimal one, such that an equivalent of Theorem 2.8 holds. This is an interesting challenge for future work. Thanks for this opportunity to clarify this point that will be mentioned explicitly.
>
> Regarding the ability to handle other activation functions: piecewise linearity of $x \mapsto g_\theta(x)$ (as in the linear and ReLU cases) is not important, and extensions e.g. to positively $p$-homogeneous activations (e.g. the squared ReLU) can be achieved [see our response to **W1b**].
>
> > **W4b** There are a few cases [...]
>
> Thank you for this suggestion, we will keep it in mind for the final version.
>
> > **Q1b** The factorization for 2-layer ReLU network requires that $\mathbb{1}(v\_j^\top x_i > 0)$ is constant. How likely does this condition hold throughout the gradient flow?
>
> This condition will *not* be preserved throughout the gradient flow, however as soon as  $\mathbb{1}(v\_j^\top x_i > 0)$ is *locally* constant we can conduct our analysis. Thus, apart from some instants along the trajectory where there are changes in these activations, the whole analysis is valid and allows to characterize the (number of independent) functions that are conserved.
>
> > **Q2b** Would it be possible to include a brief summary of what the Frobenius theorem is about? [...]
>
> Thank you for the suggestion! In the final version's supplementary material, we'll include a summary as you've recommended. Additionally, we'll provide a section that clarifies the translation of notations and vocabulary between the theorem mentioned in the given reference and our paper. When we refer to a "non-singular distribution", it implies that the dimension of the associated trace remains constant (refer to the definition of "non-singular" on page 15 of [10]). Being "involutively consistent" directly relates to our second assertion using the Lie bracket (see eq. 1.13 on page 17 of [10]). Lastly, "completely integrable" aligns with our first assertion regarding orthogonality conditions (refer to eq. 1.16 on page 23 of [10]).
>
> > **Q3b** Conservation law is also an important concept in physics [...]
>
> Our theorem is indeed related to invariance in the model (each invariance such as scaling in ReLu is associated to a conserved quantity). We will mention and clarify this connexion in the revised version. This being said, we were not able to draw a precise connexion with Noether theorem, and the settings where Noether vs Frobenius apply seem rather different.

---

> > ### Comment · Reviewer_wzMU · 2023-08-15
> >
> > Thank you for the response. I appreciate the clarification on the factorization and the Frobenius theorem. I believe the theoretical contributions are significant and have increased my score.

---

### Official Review · Reviewer_n1Hr · 2023-07-11

**Soundness:** 4 excellent
**Presentation:** 4 excellent
**Contribution:** 3 good
**Rating:** 8
**Confidence:** 3

**Summary:**

The paper studies conservation laws for deep neural networks when trained under gradient flow. Here, conservation laws refer to functions of network parameters that are invariant under gradient flow and such laws can potentially help us understand the training dynamics but constraining the manifold of parameters to a low-dimensional space and define robust symmetries of the underlying flow. This line of research had proven to be extremely fruitful for other areas of science, such as physics. The paper studies the number of such conservation laws for generic loss functions and datasets by factorizing the network function. They derive an analytical recipe to derive these numbers and provide explicit examples for simple network architectures. Finally, they provide an algorithm to compute these numbers.

**Strengths:**

The paper is mathematically engaging, well-written and the content is presented clearly.

While previous work extensively studied conservation laws, these work often restricted to specific architectures such as deep linear networks and shallow ReLU networks. This study, as far as I am aware, is the first to generically study symmetries in training dynamics for arbitrary loss functions and datasets, and can be considered as a first step towards understanding the implicit constraints coming from symmetries of various network architectures.

I believe this line of research may be impactful in many problems in DNN community such as pruning deep neural networks, principled approaches to building DNNs and more efficient training strategies.

**Weaknesses:**

1. The paper is highly technical and only provides generic mathematical tools without brining additional insights over previous findings. Space permitting, it would be helpful to show more examples that go beyond what is already known.

2. The main contribution of the paper is to derive the number of conserved quantities of a given neural network. There is no comment on the explicit constructions of such quantities in generic cases, and no application cases where these numbers can be helpful.

3. The main text lacks experiments, and the ones discussed in supplementary material are not sufficient. Explicit demonstration of conservation laws in simple neural network training might strengthen the paper.

4. The definition of the main algorithm is obscure. A step-by-step implementation might help for clarity.

**Questions:**

1. At line 120, if I am not mistaken, the fidelity function should be $f(\phi) = \sum_i \ell\left(\sum_{j,l} \varepsilon_{j, x_i} \phi_{j,k,l}  (x_i)_l, y_i\right)$, i.e. there is no summation over $k$ and the input is forgotten. Depending on how the authors feel, a subscript $f_\Omega$ can be added to emphasize locality.

2. In Eq. 3, a more generic expression should include the explicit derivative of the data fidelity function, since gradient flow may take us out the domain $\Omega$ for which $df/d\phi = \partial_\phi f$. Or are you implicitly assuming that infinitesimal gradient flow guarantees such deviations (for example in lazy learning regime)?

3. The statement at line 168 is ambiguous to me; do you mean $\nabla h \perp \chi, \forall \chi \in V$?

4. It seems to me that the effect of loss function decouples from the analysis, since the arguments only depend on $\phi$. Is it because the number of conserved quantities do not depend on the loss landscape but only the explicit form?

**Limitations:**

The applicability of the theory to practical neural networks is very restricted at the moment.

---

> ### Author Rebuttal · Authors · 2023-08-08
>
> We would like to thank the reviewer for the positive comments and constructive suggestions.
>
> **Weaknesses and Questions addressed**
>
> > **W1a** The paper is highly technical and only provides generic mathematical tools without bringing additional insights over previous findings. Space permitting, it would be helpful to show more examples that go beyond what is already known.
>
> Beyond ReLU and linear networks, which are 1-homogeneous, our results also apply to networks using any positively $p$-homogeneous activation function. Specifically, for a 2-layer NN given by $g_{\theta}(x) = U \sigma(V^\top x) $, the conserved quantities are $\|u_i\|^2 - \|v_i\|^2/p$, where $u_i$ and $v_i$ are the columns of $U$ and $V$, respectively. We plan to include this example in the supplementary material of the final version. While our framework and its generalization (refer to **W2a**) can cover more architectures, discussing them is beyond the scope of this paper.
>
> > **W2a** There is no comment on the explicit constructions of such quantities in generic cases, and no application cases where these numbers can be helpful.
>
> We believe that determining this number is a fundamental question in the study of neural networks, as it can put an end to the “quest” for potential additional laws. We disagree with the perception of a “lack of comment on explicit constructions”: we provide explicit algorithms to compute both the number of conservation laws and the laws themselves, particularly when the models are polynomial. One significant application of these conservation laws is their ability to demonstrate that, in certain cases, high-dimensional flows can be recast in lower dimensions, see Section 3.4. Another application, which we will further emphasize in the paper, aids in convergence proofs; for instance, Theorem 5 of [6] and Section 2.5 of [*], utilize balancedness conditions which are conservation laws. While it is beyond the scope of this paper, our theory could easily be applied to other architectures, including residual connections, convolutional layers, and piecewise polynomial activations.
>
> [*] On the Convergence of Gradient Descent Training for Two-layer ReLU-networks in the Mean Field Regime, S. Wojtowytsch, 2020, preprint
>
> > **W3a** The main text lacks experiments, and the ones discussed in supplementary material are not sufficient. Explicit demonstration of conservation laws in simple neural network training might strengthen the paper.
>
> Numerical illustrations of conservation laws can be found, for instance, in Figure 1 of [7] for a 2-layer linear NN, in Figure 1c of [1] for a 3-layer linear NN, and in Figure 2 of [7] for a 3-layer ReLU NN. We will include these references. In the final version of the paper, we will incorporate such a figure into the supplementary material to further emphasize our main message.
>
> > **W4a** The definition of the main algorithm is obscure. A step-by-step implementation might help for clarity.
>
> Thank you for the suggestion, we will add a pseudo-code in the final version to clarify it.
>
> > **Q1a** At line 120, if I am not mistaken, the fidelity function should be […] i.e. there is no summation over $k$ and the input is forgotten. Depending on how the authors feel, a subscript $f_{|\Omega}$ can be added to emphasize locality.
>
> Thank you, indeed there was a typo, we will correct this with a simpler (and correct!) expression using
> $ \phi_j = \phi_j(\theta) := u_jv_j^\top \in \mathbb{R}^{n \times m}$, $\phi(\theta) = (\phi_j )\_{j}$ and
> $ f(\phi) = \sum\_i \ell( \sum\_j \epsilon_{j,x_i} \phi_j x_i, y_i)$.
>
> > **Q2a** In Eq. 3, a more generic expression should include the explicit derivative of the data fidelity function, since gradient flow may take us out the domain $\Omega$ for which $df / d \phi = \partial\_{\phi} f$. Or are you implicitly assuming that infinitesimal gradient flow guarantees such deviations (for example in lazy learning regime)?
>
> Indeed it is a good idea to clarify by first writing that the gradient flow on $\mathcal{E}$ is defined as $\dot{\theta}(t) = -\nabla \mathcal{E}(\theta(t))$. Since we assume the factorization $\mathcal{E} = f \circ \phi$ on a neighborhood of $\theta_0$, for sufficiently small $t$ we deduce that the gradient flow satisfies Eq. 3. Our analysis of conservation laws is *local* to avoid considering what happens when we leave the domain.
>
> > **Q3a** The statement at line 168 is ambiguous to me; do you mean $\nabla h \perp \chi, \forall \chi \in V$?
>
> The statement that we require is stronger: it means that $\nabla h(\theta) \perp \chi(\theta), \forall \chi \in V, \forall \theta \in \Omega$. In other words, it is a pointwise assumption: at every point $\theta$, the vector $\nabla h(\theta) \in \mathbb{R}^D$ is orthogonal to the subspace $V(\theta) \subseteq \mathbb{R}^D$.
>
> > **Q4a** It seems to me that the effect of loss function decouples from the analysis, since the arguments only depend on \phi. Is it because the number of conserved quantities do not depend on the loss landscape but only the explicit form?
>
> As summarized in ll 125–129, the main idea is indeed to decouple as much as possible the study of the conserved functions from the particularities of a dataset or a given loss. This is made possible when the factorization $\mathcal{E} = f \circ \phi$ holds, and under some assumptions on the loss (see eq.7).

---

> > ### Comment · Reviewer_n1Hr · 2023-08-21
> >
> > I thank the authors for their rebuttal. I will increase my score by 1.

---

### Decision · Program_Chairs · 2023-09-21

**Decision:**

Accept (oral)

**Comment:**

A fine work proposing a more systematic approach to conservation laws for gradient flows based on concepts and tools from differential topology, in particular ones related to Frobenius theorem.

The reviewers unanimously agreed that the paper forms a solid contribution to a growing body of work in this topic, appreciating the mathematically compelling perspective it offers on the problem of characterizing continuous conservation laws.